# FSL: FEDERATED SUPERMASK LEARNING

## ABSTRACT

Federated learning (FL) allows multiple clients with (private) data to collaboratively train a common machine learning model without sharing their private training data. In-the-wild deployment of FL faces two major hurdles: robustness to poisoning attacks and communication efficiency. To address these concurrently, we propose *Federated Supermask Learning* (FSL). FSL server trains a global *subnetwork* within a *randomly initialized* neural network by aggregating local subnetworks of all collaborating clients. FSL clients share local subnetworks in the form of rankings of network edges; more useful edges have higher ranks. By sharing integer rankings, instead of float weights, FSL restricts the space available to craft effective poisoning updates, and by sharing subnetworks, FSL reduces the communication cost of training. We show theoretically and empirically that FSL is robust by design and also significantly communication efficient; all this without compromising clients' privacy. Our experiments demonstrate the superiority of FSL in real-world FL settings; in particular, **(1)** FSL achieves similar performances as state-of-the-art FedAvg with significantly lower communication costs: for CIFAR10, FSL achieves same performance as Federated Averaging while reducing communication cost by $\sim 35\%$. **(2)** FSL is substantially more robust to poisoning attacks than state-of-the-art robust aggregation algorithms.

## 1 INTRODUCTION

Federated Learning (FL) is an emerging AI technology, where mutually untrusted *clients* (e.g., Android devices) collaborate to train a shared model, called the *global model*, without explicitly sharing their local training data. FL training involves a *server* (e.g., Google server) which collects model updates from selected FL clients in each round of training, and uses them to update the global model. FL, although highly promising, faces multiple challenges (Kairouz et al., 2019; Li et al., 2020b) to its practical deployment, in particular, communication efficiency and robustness, which are the focus of our work. Privacy preservation is another major challenge to FL, but is orthogonal to our work.

We present **F**ederated **S**upermask **L**earning (FSL), a novel approach to perform FL while concurrently achieving the two goals of robustness and communication efficiency. FSL is built on a novel learning paradigm called *supermasks* (Zhou et al., 2019; Ramanujan et al., 2020), which allows it to reduce communication costs while achieving significantly higher robustness. Specifically, in FSL, clients collaborate to find a *subnetwork* within a *randomly initialized* neural network which we call the *supernetwork* (this is in contrast to conventional FL where clients collaborate to *train* a neural network). The goal of training in FSL is to collaboratively identify a supermask, which is a binary mask of 1's and 0's, that is superimposed on the random neural network (the supernetwork) to obtain the final subnetwork. The subnetwork is then used for downstream tasks, e.g., image classification, hence it is equivalent to the global model in conventional FL. Note that in entire FSL training, weights of the supernetwork *do not* change.

More specifically, each FSL client computes the importance of the edges of the supernetwork based on their local data. The importance of the edges is represented as a ranking vector. Each FSL client will use the *edge popup* algorithm (Ramanujan et al., 2020) and their data to compute their local rankings (the edge popup algorithm aims at learning which edges in a supernetwork are more important over the other edges by minimizing the loss of the subnetwork on their local data). Each client then will send their local edge ranking to the server. Finally, the FSL server uses a novel *voting* mechanism to aggregate client rankings into a supermask, which represents which edges of the random neural network (the supernetwork) will form the global subnetwork.

***Intuitions on FSL's robustness:*** In traditional FL algorithms, clients send large-dimension model updates $\in \mathbb{R}^d$ (real numbers) to the server, providing malicious clients a significant flexibility in fabricating malicious updates. By contrast, FSL clients merely share the rankings of the edges of the supernetwork, i.e., integers $\in [1, d]$, where $d$ is the size of the supernetwork. Therefore, as we will show both theoretically and empirically, FSL provides robustness by design and reduces the impact of untargeted poisoning attacks. Furthermore, unlike most existing robust FL frameworks, FSL does not require any knowledge about the percentages of malicious clients.

***Intuitions on FSL's communication efficiency:*** In FSL, the clients and the server communicate just the rankings of the edges in the supernetwork, i.e., a permutation of indices in $[1, d]$. Ranking vectors are generally significantly smaller than the global model. This, as we will show, significantly reduces the upload and download communication in FSL compared to Federated Averaging (FedAvg) (McMahan et al., 2017), where clients communicate model parameters, each of 32/64 bits.

***Empirical results:*** We experiment with three datasets in real-world heterogeneous FL settings and show that: **(1)** FSL achieves similar performance (e.g., model accuracy) as state-of-the-art FedAvg but with significantly reduced communication costs: for CIFAR10, the accuracy and communication cost per client are 85.4% and 40.2MB for FedAvg, while 85.3% and 26.2MB for FSL. **(2)** FSL is highly robust to poisoning attacks as compared to state-of-the-art robust aggregation algorithms: from 85.4% in the benign setting, 10% malicious clients reduce the accuracy of FL to 56.3% and 58.8% with Trimmed-Mean (Xie et al., 2018; Yin et al., 2018) and Multi-Krum (Blanchard et al., 2017), respectively, while FSL's performance only decreases to 79.0%.

We also compare FSL with two communication reduction methods, SignSGD (Bernstein et al., 2019) and TopK (Alistarh et al., 2018a) and show that FSL produces comparable communication costs and model accuracies. For instance, on CIFAR10, FSL, SignSGD, and TopK achieve 85.3%, 79.1%, and 82.1% test accuracy, respectively, when the corresponding communication costs (download and upload) are 26.2MB, 20.73MB, and 30.79MB. On the other hand, FSL offers a significantly superior robustness. For instance, on CIFAR10, 10% (20%) malicious clients reduce the accuracy of SignSGD to 39.7% (10.0%), but FSL's accuracy decreases to only 79.0% (69.5%). TopK is incompatible with existing robust aggregation algorithms, hence uses Average aggregation and is as vulnerable as FedAvg, especially in the real-world heterogeneous settings.

## 2 RELATED WORKS

**Supermask Learning:** Modern neural networks have a very large number of parameters. These networks are generally overparameterized (Dauphin & Bengio, 2013; Denil et al., 2013), i.e., they have more parameters than they need to perform a particular task, e.g., classification. The *lottery ticket hypothesis* (Frankle & Carbin, 2019) states that a fully-trained neural network, i.e., *supernetwork*, contains sparse *subnetworks*, i.e., subsets of all neurons in supernetwork, which can be trained from scratch (i.e., by training same initialized weights of the subnetwork) and achieve performances close to the fully trained supernetwork. The lottery ticket hypothesis allows for massive reductions in the sizes of neural networks. (Ramanujan et al., 2020) offer a complementary conjecture that an overparameterized neural network with randomly initialized weights contains subnetworks which perform as good as the fully trained network.

**Poisoning Attacks and Defenses for Federated Learning (FL):** FL involves mutually untrusting clients. Hence, a *poisoning adversary* may own or compromise some of the FL clients, called *malicious clients*, with the goal of mounting a *targeted* or *untargeted* poisoning attack. In a targeted attack, the goal is to reduce the utility of the model on specific test inputs, while in the untargeted attack, the goal is to reduce the utility for all (or most) test inputs. It is shown (Blanchard et al., 2017) that even a single malicious client can mount an effective untargeted attack on FedAvg.

In order to make FL robust to the presence of such malicious clients, the literature has designed various *robust aggregation rules (AGR)* (Blanchard et al., 2017; Mhamdi et al., 2018; Yin et al., 2018; Chang et al., 2019), which aim to remove or attenuate the updates that are more likely to be malicious according to some criterion. For instance, Multi-krum (Blanchard et al., 2017) repeatedly removes updates that are far from the geometric median of all the updates, and Trimmed-mean (Xie et al., 2018; Yin et al., 2018) removes the largest and smallest values of each update dimension and calculates the mean of the remaining values. Unfortunately, these robust AGRs are not very effective in non-convex FL settings and multiple works have demonstrated strong targeted (Wang et al., 2020;

Bhagoji et al., 2019) and untargeted attacks (Shejwalkar & Houmansadr, 2021; Fang et al., 2020) on them.

**Communication Cost of FL:** In many real-world applications of FL, it is essential to minimize the communication between FL server and clients. Especially in cross-device FL, the clients (e.g., mobile phones and wearable devices) have limited resources and communication can be a major bottleneck. There are two major types of communication reduction methods: (1) *Qunatization* methods reduce the resolution of (i.e., number of bits used to represent) each dimension of a client update. For instance, SignSGD (Bernstein et al., 2019) uses the sign (1 bit) of each dimension of model updates. (2) *Sparsification* methods propose to use only a subset of all the update dimensions. For instance, in TopK (Aji & Heafield, 2017; Alistarh et al., 2018a), only the largest K% update dimensions are sent to the server in each FL round. We note that, communication reduction methods primarily focus on and succeed at reducing upload communication (client $\rightarrow$ server), but they use the entire model in download communication (server $\rightarrow$ client).

## 3 PRELIMINARIES

### 3.1 FEDERATED LEARNING

In FL (McMahan et al., 2017; Kairouz et al., 2019; Konečný et al., 2016), $N$ clients collaborate to train a global model without directly sharing their data. In round $t$, the service provider (server) selects $n$ out of $N$ total clients and sends them the most recent global model $\theta^t$. Each client trains a local model for $E$ local epochs on their data starting from the $\theta^t$ using stochastic gradient descent (SGD). Then the client send back the calculated gradients ($\nabla_k$ for $k$th client) to the server. The server then aggregates the collected gradients and updates the global model for the next round. FL can be either cross-device or cross-silo (Kairouz et al., 2019). In cross-device FL, N is large (from few thousands to billions) and only a small fraction of clients is chosen in each FL training round, i.e., $n \ll N$. By contrast, in cross-silo FL, N is moderate (up to 100) and all clients are chosen in each round, i.e., $n = N$. In this work, we evaluate the performance of FSL and other FL baselines for cross-device FL under realistic production FL settings.

### 3.2 EDGE-POPUP ALGORITHM

The edge-popup (EP) algorithm (Ramanujan et al., 2020) is a novel optimization method to find supermasks within a large, randomly initialized neural network, i.e., a supernetwork, with performances close to the fully trained supernetwork. EP algorithm does not train the weights of the network, instead only decides the set of edges to keep and removes the rest of the edges (i.e., pop). Specifically, EP algorithm assigns a positive score to each of the edges in the supernetwork. On forward pass, it selects top k% edges with highest scores, where k is the percentage of the total number of edges in the supernetwork that will remain in the final subnetwork. On the backward pass, it updates the scores with the straight-through gradient estimator (Bengio et al., 2013). Algorithm 1 presents EP algorithm; we defer further details to Appendix D.

---

**Algorithm 1** Edge-popup (EP) algorithm: it finds a subnetwork of size $k\%$ of the entire network $\theta$

1: **Input:** number of local epochs $E$, training data $D$, initial weights $\theta^w$ and scores $\theta^s$, subnetwork size $k\%$, learning rate $\eta$
2: **for** $e \in [E]$ **do**
3: $\quad \mathcal{B} \leftarrow$ Split $D$ in $B$ batches
4: $\quad$ **for** batch $b \in [B]$ **do**
5: $\quad\quad$ EP FORWARD $(\theta^w, \theta^s, k, b)$
6: $\quad\quad \theta^s = \theta^s - \eta \nabla \ell(\theta^s; b)$
7: $\quad$ **end for**
8: **end for**
9: **return** $\theta^s$
10: **function** EP FORWARD$(\theta^w, \theta^s, k, b)$
11: $\quad m \leftarrow \text{sort}(\theta^s)$
12: $\quad t \leftarrow int((1 - k) * len(m))$
13: $\quad m[: t] = 0$
14: $\quad m[t :] = 1$
15: $\quad \theta^p = \theta^w \odot \mathbf{m}$
16: $\quad$ **return** $\theta^p(b)$
17: **end function**

---

## 4 FEDERATED SUPERMASK LEARNING: DESIGN

In this section, we provide the design of our federated supermask learning (FSL) algorithm. FSL clients collaborate (without sharing their local data) to *find a subnetwork* within a randomly initialized, untrained neural network called the *supernetwork*. Algorithm 2 describes FSL's training. Training a global model in FSL means to first find a unanimous ranking of supernetwork edges and then use the subnetwork of the top ranked edges as the final output. We detail a round of FSL training and depict it in Figure 1, where we use a supernetwork with six edges $e_{i \in [0,5]}$ to demonstrate a

single FSL round and consider three clients $C_{j \in [1,3]}$ who aim to find a subnetwork of size $k$=50% of the original supernetwork.

---

**Algorithm 2** Federated Supermask Learning (FSL)

---

1: **Input:** number of FL rounds $T$, number of local epochs $E$, number of selected users in each round $n$, seed SEED, learning rate $\eta$, subnetwork size $k\%$
2:   Server: Initialization
3: $\theta^s, \theta^w \leftarrow$ Initialize random scores and weights of global model $\theta$ using SEED
4: $R_g^1 \leftarrow$ ARGSORT$(\theta^s)$                  ▷ Sort the initial scores and obtain initial rankings
5: **for** $t \in [1, T]$ **do**
6:     $U \leftarrow$ set of $n$ randomly selected clients out of $N$ total clients
7:     **for** $u$ in $U$ **do**
8:         Clients: Calculating the ranks
9:         $\theta^s, \theta^w \leftarrow$ Initialize scores and weights using SEED
10:         $\theta^s[R_g^t] \leftarrow$ SORT$(\theta^s)$             ▷ sort the scores based on the global ranking
11:         $S \leftarrow$ Edge-PopUp$(E, D_u^{tr}, \theta^w, \theta^s, k, \eta)$     ▷ Client u uses Algorithm1 to train a supermask on its local training data
12:         $R_u^t \leftarrow$ ARGSORT$(S)$                        ▷ Ranking of the client
13:     **end for**
14:     Server: Majority Vote
15:     $R_g^{t+1} \leftarrow$ VOTE$(R_{u \in U}^t)$                         ▷ Majority vote aggregation
16: **end for**
17: **function** VOTE$(R_{\{u \in U\}})$:
18:     $V \leftarrow$ ARGSORT$(R_{\{u \in U\}})$
19:     $A \leftarrow$ SUM$(V)$
20:     **return** ARGSORT$(A)$
21: **end function**

---

### 4.1 SERVER: INITIALIZATION PHASE (ONLY FOR ROUND $t = 1$)

In the first round, the FSL server chooses a random seed SEED to generate initial random weights $\theta^w$ and scores $\theta^s$ for the global supernetwork $\theta$; note that, $\theta^w$, $\theta^s$, and SEED remain constant during the entire FSL training. Next, the FSL server shares SEED with FSL clients, who can then locally reconstruct the initial weights $\theta^w$ and scores $\theta^s$ using SEED. Figure 1-① depicts this step.

Recall that, the goal of FSL training is to find the most important edges in $\theta^w$ without changing the weights. Unless specified otherwise, both server and clients use the Singed Kaiming Constant algorithm (Ramanujan et al., 2020) to generate random weights and the Kaiming Uniform algorithm (He et al., 2015) to generate random scores. However, in Appendix C.1, we also explore the impacts of different initialization algorithms on the performance of FSL. We use the same seed to initialize weights and scores.

At the beginning, the FSL server finds the global rankings of the initial random scores (Algorithm 2 line 4), i.e., $R_g^1 =$ ARGSORT$(\theta^s)$. We define *rankings of a vector* as the indices of elements of vector when the vector is sorted from low to high, which is computed using ARGSORT function (argsort).

### 4.2 CLIENTS: CALCULATING THE RANKS (FOR EACH ROUND $t$)

In the $t^{th}$ round, FSL server randomly selects $n$ clients among total $N$ clients, and shares the global rankings $R_g^t$ with them. Each of the selected $n$ clients locally reconstructs the weights $\theta^w$'s and scores $\theta^s$'s using SEED (Algorithm 2 line 9). Then, each FSL client reorders the random scores based on the global rankings, $R_g^t$ (Algorithm 2 line 10); we depict this in Figure 1-②ⓐ.

Next, each of the $n$ clients uses reordered $\theta^s$ and finds a subnetwork within $\theta^w$ using Algorithm 1; to find a subnetwork, they use their local data and $E$ local epochs (Algorithm 2 line 11). Note that, each iteration of Algorithm 1 updates the scores $\theta^s$. Then client $u$ computes their local rankings $R_u^t$ using the final updated scores ($S$) and ARGSORT(.), and sends $R_u^t$ to the server. Figure 1-②ⓐ shows how each of the selected $n$ clients reorders the random scores using global rankings. For instance, the initial global rankings for this round are $R_g^t = [2, 3, 0, 5, 1, 4]$, meaning that edge $e_4$ should get the highest score ($s_4 = 1.2$), and edge $e_2$ should get the lowest score ($s_2 = 0.2$).

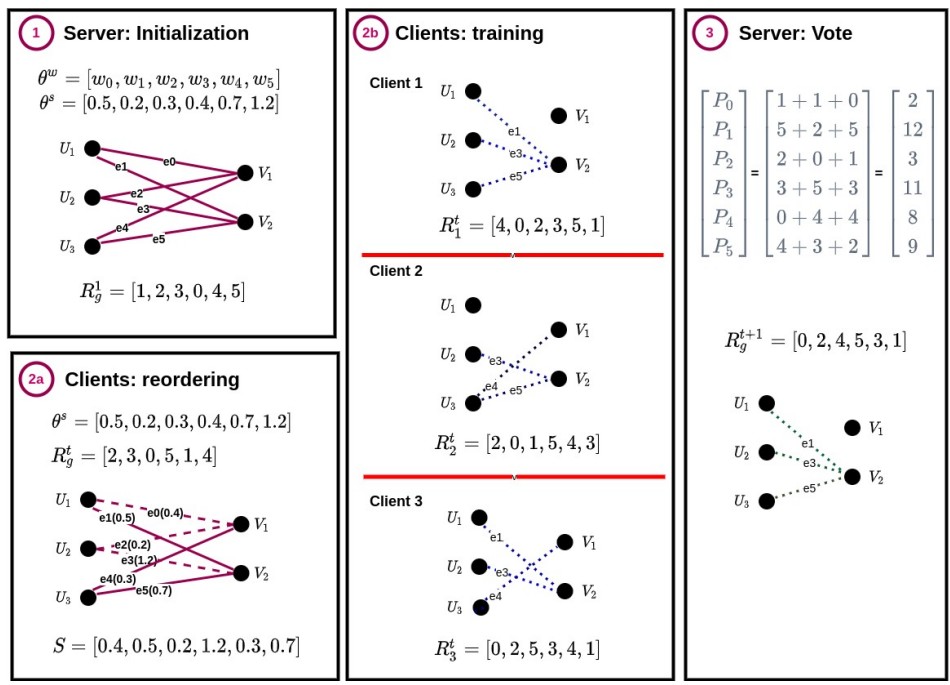

Figure 1: A single FSL round with three clients and network of 6 edges.

Figure 1-②b shows, for each client, the scores and rankings they obtained after finding their local subnetwork. For example, rankings of client $C_1$ are $R_1^t = [4, 0, 2, 3, 5, 1]$, i.e., $e_4$ is the least important and $e_1$ is the most important edge for $C_1$. Considering desired subnetwork size to be 50%, $C_1$ uses edges $\{3,5,1\}$ in their final subnetwork.

### 4.3 Server: Majority Vote (for each round $t$)

The server receives all the local rankings of the selected $n$ clients, i.e., $R_{\{u \in U\}}^t$. Then, it performs a majority vote over all the local rankings using VOTE(.) function. Note that, for client $u$, the index $i$ represents the importance of the edge $R_u^t[i]$ for $C_u$. For instance, in Figure 1-②b, rankings of $C_1$ are $R_1^t = [4, 0, 2, 3, 5, 1]$, hence the edge $e_4$ at index=0 is the least important edge for $C_1$, while the edge $e_1$ at index=5 is the most important edge. Consequently, VOTE(.) function assigns reputation=0 to edge $e_4$, reputation=1 to $e_0$, reputation=2 to $e_2$, and so on. In other words, for rankings $R_u^t$ of $C_u$ and edge $e_i$, VOTE(.) computes the reputation of $e_i$ as its index in $R_u^t$. Finally, VOTE(.) computes the total reputation of $e_i$ as the sum of reputations from each of the local rankings. In Figure 1-②b, reputations of $e_0$ are 1 in $R_1^t$, 1 in $R_2^t$, and 0 in $R_3^t$, hence, the total reputation of $e_0$ is 2. We depict this in Figure 1-③; here, the final total reputations for edges $e_{\{i \in [0,5]\}}$ are $A = [2, 12, 3, 11, 8, 9]$. Finally, the server computes global rankings $R_g^{t+1}$ to use for round $t + 1$ by sorting the final total reputations of all edges, i.e., $R_g^{t+1} = \text{ARGSORT}(A)$.

Note that, *all FSL operations that involve sorting, reordering, and voting are performed in a layer-wise manner*. For instance, the server computes global rankings $R_g^t$ in round $t$ for each layer separately, and consequently, the clients selected in round $t$ reorder their local randomly generated scores $\theta^s$ for each layer separately.

## 5 Federated Supermask Learning: Salient Features

In this section, we discuss the two salient features of FSL that are instrumental for any distributed learning algorithm to be practical: *robustness to poisoning attacks* and *communication efficiency*.

### 5.1 Robustness of FSL to Poisoning Attacks

FSL is a distributed learning algorithm with mutually untrusting clients. Hence, a *poisoning adversary* may own or compromise some of FSL clients, called *malicious clients*, and mount a *targeted* or *untargeted* poisoning attack. In our work, we consider the untargeted attacks as they are more severe than targeted attacks and can cause denial-of-service for all collaborating clients (Shejwalkar et al., 2021) and show that FSL is secure against such poisoning attacks by design.

**Intuition on FSL's robustness:** Existing FL algorithms, including robust FL algorithms, are shown to be vulnerable to targeted and untargeted poisoning attacks (Shejwalkar et al., 2021) where malicious clients corrupt the global model by sharing malicious model updates. One of the key reasons behind the susceptibility of existing algorithms is that their model updates can have arbitrary values. For instance, to manipulate vanilla FedAvg, malicious clients send very large updates (Blanchard et al., 2017), and to manipulate Multi-krum and Trimmed-mean, (Fang et al., 2020; Shejwalkar & Houmansadr, 2021) propose to perturb a benign update in a specific malicious direction. On the other hand, in FSL, clients must send a permutation of indices $\in [1, n_\ell]$ for each layer. Hence, FSL significantly reduces the space of the possible malicious updates that an adversary can craft. Majority voting in FSL further reduces the chances of successful attack. Intuitively, this makes FSL design robust to poisoning attacks. Below, we make this intuition more concrete.

**The worst-case untargeted poisoning attack on FSL:** Here, the poisoning adversary aims to reduce the accuracy of the final global FSL subnetwork on most test inputs. To achieve this, the adversary should replace the high ranked edges with low ranked edges in the final subnetwork. For the worst-case analysis of FSL, we assume a very strong adversary (i.e., threat model): 1) each of the malicious clients has some data from benign distribution; 2) malicious clients know the entire FSL algorithm and its parameters; 3) malicious clients can collude. Under this threat model we design a worst case attack on FSL (Algorithm 3 in Appendix A.1), which executes as follows: First, all malicious clients compute rankings on their benign data and use VOTE(.) algorithm to compute an aggregate rankings. Finally, each of the malicious clients uses the reverse of the aggregate rankings to share with the FSL server in given round. The adversary should invert the rankings layer-wise as the FSL server will aggregate the local rankings per layer too, and it is not possible to mount a model-wise attack.

Now we justify why the attack in Algorithm 3 is the worst case attack on FSL for the strong threat model we consider. Note that, FSL aggregation, i.e., VOTE(.), computes the reputations using clients' rankings and sums the reputations of each network edge. Therefore, the strongest poisoning attack would want to reduce the reputation of good edges. This can be achieved following the aforementioned procedure of Algorithm 3 to reverse the rankings computed using benign data.

**Theoretical analysis of robustness of FSL algorithm:**
In this section, we prove an upper bound on the failure probability of robustness of FSL, i.e., the probability that a good edge will be removed from the final subnetwork when malicious clients mount the worst case attack.

Following the work of Bernstein et al. (2019), we make two assumptions in order to facilitate a concrete robustness analysis of FSL: a) each malicious client has access only to its own data, and b) we consider a simpler VOTE(.) function, where the FSL server puts an edge $e_i$ in the final subnetwork if more than half of the clients have $e_i$ (a good edge) in their local subnetworks. In other words, the rankings that each client sends to the server is just a bit mask showing that each edge should or should not be in the final subnetwork. The server makes a majority vote on the bit masks, and if an edge has more than half votes, it will be in the global subnetwork. Our VOTE(.) mecha-

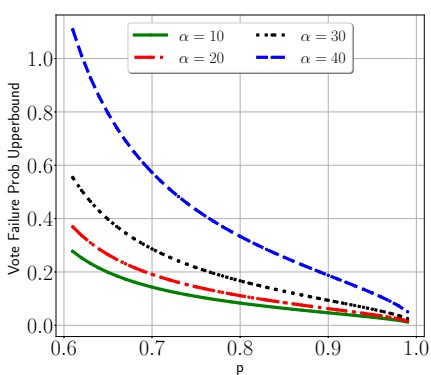

Figure 2: Upper bound on the failure probability of VOTE(.) function in FSL.

nism has more strict robustness criterion, as it uses more nuanced reputations of edges instead of bit masks. Hence, the upper bound on failure probability in this section also applies to the FSL VOTE(.) function.

The probability that our voting system fails is the probability that more than half of the votes do not include $e_i$ in their subnetworks. The upper bound on the probability of failure would be $1/2 \sqrt{\frac{np(1-p)}{(n(p+\alpha(1-2p)-1/2))^2}}$, where $n$ is the number of clients being processed, $p$ shows the probability that a benign client puts $e_i$ in their top ranks, and $\alpha$ is the fraction of malicious clients. Due to space limitations, we defer the detailed proof to Appendix A.2. Figure 2 shows the upper bound on the failure of VOTE(.) for different values of $\alpha$ and $p$. We note that, the higher the probability $p$, the higher the robustness of FSL.

## 5.2 FSL's COMMUNICATION COSTS

In FL, and especially in the cross-device setting, clients have limited communication bandwidth. Hence, FL algorithms must be communication efficient. We discuss here the communication cost of FSL algorithm. In the first round, the FSL server only sends one seed of 32 bits to all the FSL clients, so they can construct the random weights ($\theta^w$) and scores ($\theta^s$). In a round $t$, each of selected FSL clients receives the global rankings $R_g^t$ and sends back their local rankings $R_u^t$. The rankings are a permutation of the indices of the edges in each layer, i.e., of $[0, n_\ell - 1] \forall \ell \in [L]$ where $L$ is the number of layers and $n_\ell$ is the number of parameters in $\ell$th layer.

We use the naive approach to communicate layer-wise rankings, where each FSL client exchanges a total of $\sum_{\ell \in [L]} n_\ell \times \log(n_\ell)$ bits. Because, for the layer $\ell$, the client receives and sends $n_\ell$ ranks where each one is encoded with $\log(n_\ell)$ bits. On the other hand, a client exchanges $\sum_{\ell \in [L]} n_\ell \times 32$ bits in FedAvg, when 32 bits are used to represent each of $n_\ell$ weights in layer $\ell$. In Appendix E, we compare theoretical communication costs of various FL algorithms.

**Sparse-FSL:** Here, we propose Sparse-FSL, a simple extension of FSL to further reduce the communication cost. In Sparse-FLS, a client sends only the most important ranks of their local rankings to the server for aggregation. For instance, in Figure 1, client $C_1$ sends $R_1^t = [4, 0, 2, 3, 5, 1]$ in case of FSL. But in sparse-FSL, with sparsity set to 50%, client $C_1$ sends just the top 3 rankings, i.e., sends $R_1'^t = [3, 5, 1]$. For each client, the sparse-FSL server assumes 0 reputation for all of the edges not included in the client's rankings, i.e., in Figure 1, sparse-FSL server will assign reputation=0 for edges $e_4$, $e_0$, and $e_2$. Then the server uses VOTE(.) to compute total reputations of all edges and then sort them to obtain the final aggregate global rankings, i.e., $R_g^{t+1}$, to use for subsequent rounds. We observe in out experiments, that sparse-FSL performs very close to FSL, even with sparsity as low as 10%, while also significantly reducing the communication cost. Due to space limitation, we defer the communication cost comparison of FSL with FedAvg, SingSGD, and LotteryFL (Li et al., 2020a) to Appendix E.

## 6 EXPERIMENTS

In this section, we investigate the utility, robustness, and communication cost of our FSL framework. We use MNIST, CIFAR10, and FEMNIST data and distribute them in non-iid fashion (using Dirichlet distribution) among 1000, 1000, and 3400 clients respectively. At the end of the training, we calculate the test accuracy of all the clients on the final global model, and we report the mean and standard deviation of all clients' test accuracies in our experiments. We provide further details of experimental setup in Appendix B. In addition to FSL, we also evaluate Sparse-FSL in different settings. We use SFSL top x% to denote a Sparse-FSL clients who sends top x% of ranks in each round.

### 6.1 COMMUNICATION COST ANALYSIS

In FSL, both clients and server communicate just the edge ranks instead of weight parameters. Thus, FSL reduces both upload and download communication cost. Table 1 illustrates the utility, i.e., the accuracy on test data, and communication cost of FSL and state-of-the-art quantization, i.e., SignSGD (Bernstein et al., 2019), and sparsification, i.e., TopK (Alistarh et al., 2018b; Aji & Heafield, 2017) communication-reduction methods.

**FSL versus SignSGD:** We note that, FSL is significantly more accurate than SignSGD. For instance, on CIFAR10, distributed non-iid among 1000 clients, FSL achieves 85.3% while SignSGD achieves 79.1% , or on FEMNIST, FSL achieves 84.2% while SignSGD achieves 79.3%. This is because, FSL clients send more nuanced information via rankings of their subnetworks compared to SignSGD, where clients just send the signs of their model updates.

SignSGD in FL reduces only the upload communication, but for efficiency reasons, the server sends all of the weight parameters (each of 32 bits) to the newly selected clients. Hence, SignSGD has very efficient upload communication, but very inefficient download communication. For instance, on CIFAR10, for both upload and download, FSL achieves 13.1MB each while SignSGD achieves 0.63MB and 20.1MB, respectively.

**FSL versus TopK:** We compare FSL with TopK with $K \in \{10, 50\}\%$. FSL is more accurate than Topk for MNIST and CIFAR10: on CIFAR10, FSL accuracy is 85.3%, while TopK accuracies are 82.1% and 77.8% with $K$=50% and $K$=10%, respectively. Similar to SignSGD, Topk is more efficiently reduces upload cost, but has very high download communication cost. Therefore, the

Table 1: Comparing the utility (test accuracy) and communication cost of FedAvg, FSL (in **bold**), SignSGD and, TopK and Sparse-FSL (SFSL) with different percentages of sparsity (in **bold**).

| Dataset | Algorithm | Accuracy (STD) | Upload (MB) | Download (MB) |
|---|---|---|---|---|
| MNIST + LeNet 1000 clients | FedAvg | 98.8 (3.1) | 6.20 | 6.20 |
| | **FSL** | **98.8 (3.2)** | **4.05** | **4.05** |
| | **SFSL Top 50%** | **98.2 (3.8)** | **2.03** | **4.05** |
| | **SFSL Top 10%** | **89.5 (9.2)** | **0.40** | **4.05** |
| | SignSGD | 97.2 (4.6) | 0.19 | 6.20 |
| | TopK 50% | 98.8 (3.2) | 3.29 | 6.20 |
| | TopK 10% | 98.7 (3.2) | 0.81 | 6.20 |
| CIFAR10 + Conv8 1000 clients | FedAvg | 85.4 (11.2) | 20.1 | 20.1 |
| | **FSL** | **85.3 (11.3)** | **13.1** | **13.1** |
| | **SFSL Top 50%** | **77.6 (13.0)** | **6.5** | **13.1** |
| | **SFSL Top 10%** | **27.5 (14.4)** | **1.3** | **13.1** |
| | SignSGD | 79.1 (13.6) | 0.63 | 20.1 |
| | TopK 50% | 82.1 (11.8) | 10.69 | 20.1 |
| | TopK 10% | 77.8 (13.0) | 2.64 | 20.1 |
| FEMNIST + LeNet 3400 clients | FedAvg | 85.8 (10.2) | 6.23 | 6.23 |
| | **FSL** | **84.2 (10.7)** | **4.06** | **4.06** |
| | **SFSL Top 50%** | **75.2 (12.7)** | **2.03** | **4.06** |
| | **SFSL Top 10%** | **59.2 (15.0)** | **0.40** | **4.06** |
| | SignSGD | 79.3 (12.4) | 0.19 | 6.23 |
| | TopK 50% | 85.7 (9.9) | 3.31 | 6.23 |
| | TopK 10% | 85.5 (10.0) | 0.81 | 6.23 |

combined upload and download communication cost per client per round is 26.2MB for FSL and 30.79MB for TopK with $K$=50%, and TopK still has worse performance.

**Communication cost reduction due to Sparse-FSL (SFSL):** We now evaluate SFSL explained in Section 5.2. In SFSL with top 50% ranks, clients send the top 50% of their ranks to the server, which reduces the upload bandwidth consumption by half. Please note that the download cost of SFSL is the same as FSL since the FSL server should send all the global rankings to the selected clients in each round. We note from Table 1 that, by sending fewer ranks, SFSL reduces upload communication at a small cost of performance. For instance, on CIFAR10, SFLS with top 50% reduces the upload communication by 50% at the cost reducing accuracy from 85.4% to 77.6%.

## 6.2 SECURITY ANALYSIS

We compare FSL with state-of-the-art robust aggregation rules (AGRs): Mkrum (Blanchard et al., 2017), and Trimmed-mean (Xie et al., 2018; Yin et al., 2018). Table 2 gives the performances of robust AGRs, SignSGD, and FSL with different percentages of malicious clients. Here, we make a rather impractical assumption in favor of the robust AGRs: we assume that the server knows the exact % of malicious clients in each FL round. FSL does not require this knowledge.

**FSL achieves higher robustness than state-of-the-art robust AGRs:** We note from Table 2 that, FSL is more robust to the presence of malicious clients who try to poison the global model compared to Multi-Krum, Trimmed-mean, and SignSGD for both 10% and 20% malicious clients rates. For instance, on CIFAR10, 10% malicious clients can decrease the accuracy of FL models to 56.3%, 58.8%, and 39.8% for Trimmed-mean, Multi-Krum, and SignSGD respectively; 20% malicious clients can decrease the accuracy of the FL models to 20.5%, 25.6%, 10.0% for Trimmed-mean, Multi-Krum, and SignSGD respectively. On the other hand, FSL performance decreases to 79.0% and 69.5% for 10% and 20% attacking ratio respectively.

We make similar observations for MNIST and FEMNIST datasets: for FEMNIST, 10% (20%) malicious clients reduce accuracy of the global model from 85.8% to 72.7% (56.2%) for Trimmed-Mean, to 80.9% (23.7%) for Multi-krum, and 76.7% (55.1%) for SignSGD, while FSL accuracy decreases to 83.0% (65.8%). We omit evaluating TopK, because even a single malicious client (Blanchard et al., 2017) can jeopardize its accuracy.

Table 2: Comparing the robustness of various FL algorithms: FSL and SFSL (in **bold**) outperform state-of-the-art robust AGRs and SignSGD against strongest of untargeted poisoning attacks.

| Dataset | AGR | No malicious | 10% malicious | 20% malicious |
|---|---|---|---|---|
| MNIST + LeNet 1000 clients | FedAvg | 98.8 (3.2) | 10.0 (10.0) | 10.0 (10.0) |
| | Trimmed-mean | 98.8 (3.2) | 95.1 (7.7) | 87.6 (9.5) |
| | Multi-krum | 98.8 (3.2) | 98.6 (3.3) | 97.9 (4.1) |
| | SignSGD | 97.2 (4.6) | 96.6 (5.0) | 96.2 (5.6) |
| | **FSL** | **98.8 (3.1)** | **98.8 (3.1)** | **98.7 (3.3)** |
| | **SFSL Top 50%** | **98.2 (3.8)** | **97.04 (4.4)** | **95.1 (7.8)** |
| CIFAR10 + Conv8 1000 clients | FedAvg | 85.4 (11.2) | 10.0 (10.1) | 10.0 (10.1) |
| | Trimmed-mean | 84.9 (11.0) | 56.3 (16.0) | 20.5 (13.2) |
| | Multi-krum | 84.7 (11.3) | 58.8 (15.8) | 25.6 (14.4) |
| | SignSGD | 79.1 (12.8) | 39.7 (15.9) | 10.0 (10.1) |
| | **FSL** | **85.3 (11.3)** | **79.0 (12.4)** | **69.5 (14.8)** |
| | **SFSL Top 50%** | **77.6 (13.0)** | **41.7 (15.4)** | **39.7 (15.2)** |
| FEMNIST + LeNet 3400 clients | FedAvg | 85.8 (10.2) | 6.3 (5.8) | 6.3 (5.8) |
| | Trimmed-mean | 85.2 (11.0) | 72.7 (15.7) | 56.2 (20.3) |
| | Multi-krum | 85.2 (10.9) | 80.9 (12.2) | 23.7 (12.8) |
| | SignSGD | 79.3 (12.4) | 76.7 (13.2) | 55.1 (14.9) |
| | **FSL** | **84.2 (10.7)** | **83.0 (10.9)** | **65.8 (17.8)** |
| | **SFSL Top 50%** | **75.2 (12.7)** | **70.5 (14.4)** | **60.39 (14.8)** |

## 6.3 MISCELLANEOUS DISCUSSIONS

Due to space limitations, we defer a detailed discussion of ablation studies of FSL to Appendix C and below give their important takeaways.

**Initialization matters in FSL:** In FSL, the weight parameters are randomly initialized at the start and remain fixed throughout the training. An appropriate initialization is instrumental to the success of FSL, since the clients are trying to find the most important weight parameters. We study efficacy of three initializing strategies that use three different distributions: Glorot Normal, Kaiming Normal, and Singed Kaiming Constant. Table 5 shows the results. We observe from Table 5 that, Singed Kaiming Constant initialization achieves the best results that are closest to FedAvg.

**Varying the sparsity of edge-popup algorithm in FSL:** Figure 4 illustrates how the performance of FSL varies with the sizes of local subnetworks that the clients share with the server. In other words, when we vary the sparsity $k\%$ of edge popup algorithm during local subnetwork search $k \in [10, 20, 30, 40, 50, 60, 70, 80, 90]\%$. Interestingly we note that, FSL performs the worst when clients use all ($k$=100%) or none ($k$=0%) of the edges. This is because, it is difficult to find a subnetwork with small number of edges. While using all of the edge essentially results in using a random neural network. As we can see FSL with $k \in [40, 70]\%$, gives the best performances for all the three datasets. Hence, we set $k$=50% by default in our experiments.

## 7 CONCLUSIONS

We designed a novel collaborative learning algorithm, called *Federated Supermask Learning* (FSL), to address the issues of robustness to poisoning and communication efficiency in existing FL algorithms. We argue that a core reason for the susceptibility of existing FL algorithms to poisoning is the use of arbitrary values in their model updates. Hence, in FSL, we use ranks of edges of a randomly initialized neural network contributed by collaborating clients to find a global ranking and then use a subnetwork based only on the top edges. Use of rankings in a fixed range restricts the space available to poisoning adversaries to craft malicious updates, and also allows FSL to use sophisticated communication reduction methods. We show, both theoretically and empirically, that ranking based collaborative learning can effectively mitigate the robustness issue as well as reduce the communication costs involved.

## 8 REPRODUCIBILITY STATEMENT

We have attached our implementation code and experiment notebooks as supplementary materials. There are separate experiment notebooks showing the process of FSL training for MNIST, CIFAR10, and FEMNIST (results that we used in Table 1 and Table 2). We have explained all the hyperparameters including optimizer, learning rate, momentum, weight decay, and batch size in Appendix B.1 for both FSL and FedAvg (including using robust AGRs) settings. We also explained the model architectures we used in our experiments in Table 4 explaining different layers with number of parameters inside each. We also explained how we distributed MNIST and CIFAR10 samples among 1,000 clients using Dirichlet distribution to make them non-iid in Appendix B.3.

We propose *Federated Supermask Learning* (FSL) where clients find a *subnetwork* within a randomly initialized neural network, *without* training the weights of the network. Our extensive evaluation demonstrated FSL can provide better communication cost and more robustness to untargeted poisoning attacks compared to existing FL compressors and Byzantine-robust aggregation rules.

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

# A MISSING DETAILS OF ROBUSTNESS OF FSL

## A.1 FSL WORST CASE POISONING ATTACK ALGORITHM

Algorithm 3 shows the rankings poisoning attack explained in Section 5.

---

**Algorithm 3** FSL Poisoning

---

1: **Input:** number of malicious clients $M$, number of malicious local epochs $E'$, seed SEED, global ranking $R_g^t$, learning rate $\eta$, subnetwork size $k\%$
2: **function** CRAFTMALICIOUSUPDATE($M$, SEED, $R_g^t$, $E'$, $\eta$, $k$):
3:     **for** $mu \in [M]$ **do**                                       $\triangleright$ For all the malicious clients
4:         Malicious Client Executes:
5:         $\theta^s, \theta^w \leftarrow$ Initialize scores and weights using SEED
6:         $\theta^s[R_g^t] \leftarrow$ SORT($\theta^s$)
7:         $S \leftarrow$ Edge-PopUp($E'$, $D_u^{tr}$, $\theta^w$, $\theta^s$, $k$, $\eta$)
8:         $R_{mu}^t \leftarrow$ ARGSORT($S$)                       $\triangleright$ Ranking of the malicious client
9:     **end for**
10:     Aggregation:
11:     $R_m^t \leftarrow$ VOTE($R_{mu \in [M]}^t$)                   $\triangleright$ Majority vote aggregation
12:     **return** REVERSE($R_m^t$)                      $\triangleright$ reverse the ranking
13: **end function**

---

## A.2 THEORETICAL ANALYSIS OF ROBUSTNESS OF FSL

In this section, we detail the proof of robustness of FSL. In other words, we prove an upper bound on the failure probability of robustness of FSL, i.e., the probability that a good edge will be removed from the final subnetwork when malicious clients mount the worst case attack. Inspired from SignSGD (Bernstein et al., 2019), for this proof, We assume a simpler VOTE(.) function where if more than half of the clients add an edge $e_i$ to their subnetworks, then the FSL server adds it to the final global subnetwork. We also assume that the malicious clients cannot collude in our proof.

Assume that edge $e_i$ is a good edge, i.e., having $e_i$ in the final subnetwork improves the performance of the final subnetwork. Let Z be the random variable that represents the number of clients who vote for the edge $e_i$ to be in the final subnetwork, i.e., the number of clients whose local subnetwork of size $k\%$ of the entire supernetwork (Algorithm 2 line 11) contains $e_i$. Therefore, $Z \in [0, n]$ where $n$ is the number of clients being processed in a given FSL round.

Let G and B be the random variable that represent the number of benign and malicious clients that vote for edge $e_i$, respectively; the malicious clients inadvertently exclude the good edge $e_i$ in their local subnetwork based on their benign training data.

There are total of $\alpha n$ malicious clients, where $\alpha$ is the fraction of malicious clients that B of them decides that $e_i$ is a bad edge and should not be removed. Each of the malicious clients computes the subnetwork on its own benign training data, so B of them do not conclude that $e_i$ is a good edge. Hence, $Z = G + B$. We can say that G and B have binomial distribution , i.e., $G \sim$ binomial($[(1-\alpha)n, p]$ and $B \sim$ binomial($[\alpha n, 1-p]$ where $p$ is the probability that a benign client has this edge in their local subnetwork and $\alpha$ is the fraction of malicious clients. Note that the probability that our voting in simplified FSL fails is $P[\text{failure}] = P[Z <= \frac{n}{2}]$, i.e., when more than half of the clients vote against $e_i$, i.e., they do not include $e_i$ in their local subnetworks. We can find the mean and variance of Z as follows:

$$E[Z] = (1-\alpha)np + \alpha n(1-p) \tag{1}$$

$$Var[Z] = (1-\alpha)np(1-p) + \alpha np(1-p) = np(1-p) \tag{2}$$

Cantelli (1929) provides an inequality where for a random variable X with mean $\mu$ and variance $\sigma^2$ we have $P[\mu - X >= \lambda] <= \frac{1}{1 + \frac{\lambda^2}{\sigma^2}}$. Using this inequality, we can write:

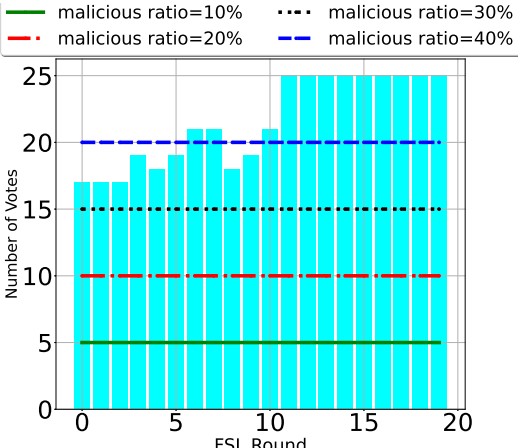

Figure 3: Empirical robustness bounds of FSL. Bars are showing the number of votes for a particular edge in first layer of LeNet (trained on MNIST) with selecting 25 clients in each round and no presence of malicious client. The adversary tries to flip one good edge from the global model. The horizontal lines are showing the thresholds for each malicious ratio that if the number of votes is less than them, the adversary can change the decision about this edge.

$$P[Z <= \frac{n}{2}] = P[E[Z] - Z >= E[Z] - n/2] <= \frac{1}{1 + \frac{(E[z] - n/2)^2}{var[Z]}} \qquad (3)$$

because $1 + x^2 >= 2x$, we have:

$$P[Z <= \frac{n}{2}] <= 1/2\sqrt{\frac{Var[Z]}{(E[Z] - n/2)^2}} = 1/2\sqrt{\frac{np(1-p)}{(np - \alpha np + \alpha n - \alpha np - n/2)^2}} \qquad (4)$$

$$= 1/2\sqrt{\frac{np(1-p)}{(n(p + \alpha(1 - 2p) - 1/2))^2}}$$

What this means is that the probability that the simplified VOTE(.) fails is upper bounded as in (4). We show the effect of the different values of $\alpha$ and $p$ in Figure 2. We can see from Figure 2, if the benign clients can train better supermasks (better chance that a good edge ended in their subnetwork), the probability that the attackers succeed is lower (more robustness). VOTE(.) in FSL (Section 4.3) is more sophisticated and puts more constraints on the malicious clients, hence the about upper bound also applies to FSL.

We show the theoretical relationship between upper bound on the failure of VOTE(.) for different values of malicious rate ($\alpha$) in Figure 2. To validate our theoretical bounds, we measure the least number of malicious clients that the adversary needs to control to remove a good edge from the global subnetwork in Figure 3. In this figure, we report the votes of one particular edge in the first layer of LeNet (trained on MNIST) where there are 288 edges in the first layer (Table B.1 shows the number of edges in each layer). We consider this edge as a good edge since we observe that it would be in the final global subnetwork (at FSL round 20) if there were no malicious clients. In this figure, the bars are showing the number of votes this edge received to be in the global subnetwork for that FSL round. The horizontal lines are showing the thresholds that if the number of votes is less than them, the adversary can change the decision about this good edge. We are selecting 25 clients in each round, so we consider $2 \times \alpha \times 25$ for thresholds. For instance, when we assume there are 20% malicious clients (on average there are 5 malicious among 25 selected clients) that means that 5 votes of benign votes decreases and 5 votes is added to malicious votes, so the threshold would be 10.

Table 3: Model architectures. We use identical architecture to those Ramanujan et al. (2020); Wortsman et al. (2020) used.

| Architecture | Layer Name | Number of parameters |
|---|---|---|
| LeNet + MNIST (Wortsman et al., 2020) | Convolution(32) + Relu | 288 |
| | Convolution(64) + Relu | 18432 |
| | MaxPool(2x2) | - |
| | FC(128) + Relu | 1605632 |
| | FC(10) | 1280 |
| Conv8 + CIFAR10 (Ramanujan et al., 2020) | Convolution(64) + Relu | 1728 |
| | Convolution(64) + Relu | 36864 |
| | MaxPool(2x2) | - |
| | Convolution(128) + Relu | 73728 |
| | Convolution(128) + Relu | 147456 |
| | MaxPool(2x2) | - |
| | Convolution(256) + Relu | 294912 |
| | Convolution(256) + Relu | 589824 |
| | MaxPool(2x2) | - |
| | Convolution(512) + Relu | 1179648 |
| | Convolution(512) + Relu | 2359296 |
| | MaxPool(2x2) | - |
| | FC(256) + Relu | 524288 |
| | FC(256) + Relu | 65536 |
| | FC(10) | 2560 |
| LeNet + FEMNIST (Wortsman et al., 2020) | Convolution(32) + Relu | 288 |
| | Convolution(64) + Relu | 18432 |
| | MaxPool(2x2) | - |
| | FC(128) + Relu | 1605632 |
| | FC(62) | 7936 |

# B    MISSING DETAILS OF EXPERIMENTAL SETUP

## B.1    DATASETS AND MODEL ARCHITECTURES

**MNIST** is a 10-class class-balanced classification task with 70,000 gray-scale images, each of size $28 \times 28$. We experiment with LeNet architecture given in Table 4. For local training in each FSL/FL round, each client uses 2 epochs. For training weights (experiments with FedAvg, SignSGD, TopK), we use SGD optimizer with learning rate of 0.01, momentum of 0.9, weight decay of 1e-4, and batch size 8. For training supermasks (experiments with FSL), we use SGD with learning rate of 0.4, momentum 0.9, weight decay 1e-4, and batch size 8.

**CIFAR10** (Krizhevsky & Hinton, 2009) is a 10-class classification task with 60,000 RGB images (50,000 for training and 10,000 for testing), each of size $32 \times 32$. We experiment with a VGG-like architecture given in Table 4, which is identical to what Ramanujan et al. (2020) used. For local training in each FSL/FL round, each client uses 5 epochs. For training weights (experiments with FedAvg, SignSGD, TopK), we use SGD with learning rate of 0.1, momentum of 0.9, weight decay of 1e-4, and batch size of 8. For training supermasks (experiments with FSL), we optimize SGD with learning rate of 0.4, momentum of 0.9, weight decay of 1e-4, and batch size of 8.

**FEMNIST** (Caldas et al., 2018; Cohen et al., 2017) is a character recognition classification task with 3,400 clients, 62 classes (52 for upper and lower case letters and 10 for digits), and 671,585 gray-scale images. Each client has data of their own handwritten digits or letters. We use LeNet architecture given in Table 4. For local training in each FSL/FL round, each client uses 2 epochs. For training weights (experiments with FedAvg, SignSGD, TopK), we use SGD with learning rate of 0.15, momentum of 0.9, weight decay of 1e-4, and batch size of 10. For training supermasks (experiments with FSL), we optimize SGD with learning rate of 0.2, momentum of 0.9, weight decay of 1e-4, and batch size of 10.

## B.2  HYPERPARAMETERS TUNING

We optimize the hyperparameters based on FSL and other baselines independently. The hyperparameters that we used in our experiments are tuned in scenario with no malicious clients. Table B.1 shows the performance of FSL and other baselines on CIFAR10 (distributed over 1000 users using Dirichlet distribution) for different values of hyperparameters when there are 10% malicious clients among the clients. This table shows the robustness of FSL still persists even if we change the hyperparameters. We reported mean of accuracies and standard deviation of accuracies for all the clients at the final FSL round.

### B.3  NON-IID DATA DISTRIBUTION

**Using Dirichlet Distribution:**  Considering the heterogeneous data in the real-word cross-device FL, similar to previous works (Reddi et al., 2020; Hsu et al., 2019), we distribute MNIST and CIFAR10 among 1,000 clients in a non-iid fashion using Dirichlet distribution with parameter $\beta = 1$. Note that increasing $\beta$ results in more iid datasets. Next, we split datasets of each client into training (80%) and test (20%). At the end of the FL rounds, we calculate the test accuracy of each client for its test data, and we report the average of test accuracies of all the clients. We run all the experiments for 2000 global rounds of FSL and FL and select 25 clients in each round.

**Assigning only two classes to each client:**  McMahan et al. (2017) used a more extreme heterogeneous data assignment. For assignment of MNIST and CIFAR10 among 1000 clients using this Non-iid-ness method, we sort all the training and validation data inside MNIST and CIFAR10, then partition them into 2000 shards. This means that each shards of training MNIST has 30 images and each CIFAR10 shard has 25 images. Then we assign two random shards to each client resulting in each client have at most data of two classes, and in CIFAR10 experiments, each client has 50 training images, and 10 test images, and in MNIST experiments, each client has 60 training images and 10 test images. We only use this assignment in Section C.3.

### B.4  BASELINE FL ALGORITHMS

**Federated averaging**  In non-adversarial FL settings, i.e., without any malicious clients, the dimension-wise Average (FedAvg) (Konečný et al., 2016; McMahan et al., 2017) is an effective AGR. In fact, due to its efficiency, Average is the only AGR implemented by FL applications in practice (Ludwig et al., 2020; Paulik et al., 2021).

**SignSGD**  is a quantization method used in distributed learning to compress each dimension of updates into 1 bit instead of 32 or 64 bits. To achieve this, in SignSGD (Bernstein et al., 2019) the clients only send the sign of the gradient updates to the server, and the server runs a majority vote on them. SignSGD is designed for distributed learning where all the clients participate in each round, so all the clients are aware of the most updated weight parameters of the global model. However, using SignSGD in FL just provides benefit in upload bandwidth, but to achieve good overall performance of the global model, the server should send all the weight parameters (each of 32 bits) to the newly selected clients in each round. This makes SignSGD very efficient in upload cost, but it is as inefficient as FedAvg in download.

**TopK**  is a sparsification method used in distributed learning that transmits only a few elements in each model update to the server. In TopK (Aji & Heafield, 2017; Alistarh et al., 2018a), the clients first sort the absolute values of their local gradient updates, and send the Top K% largest gradients update dimensions to the server for aggregation. TopK suffers from the same problem as SignSGD: for performance reasons, the server should send the entire updated model weights to the new selected clients.

### B.5  MODEL POISONING ATTACK FOR ROBUSTNESS EVALUATIONS

To evaluate robustness of various FL algorithms, we use state-of-the-art model poisoning attack proposed by  Shejwalkar & Houmansadr (2021) in our robustness experiments. The attack proposes a general FL poisoning framework and then tailors it to specific FL settings. It first computes an average $\nabla^b$ of the available benign updates and perturbs it in a *dynamic, data-dependent malicious direction* $\omega$ to compute the final poisoned update $\nabla' = \nabla^b + \gamma\omega$. DYN-OPT finds the largest $\gamma$ that

Table 4: Performance of FSL with different hyperparameters trained on CIFAR10 (distributed over 1000 clients using Dirichlet distribution).

| Method | hyperparameter | value | Test Accuracy with 10% malicious |
|---|---|---|---|
| FSL | batch size | 6 | 78.4 (12.6) |
| | | 8 | 79.0 (12.4) |
| | | 16 | 76.4 (13.6) |
| | local epochs | 2 | 79.8 (12.2) |
| | | 5 | 79.0 (12.4) |
| | | 10 | 78.2 (12.6) |
| | learning rate | 0.1 | 73.5 (13.4) |
| | | 0.2 | 82.4 (12.1) |
| | | 0.3 | 83.11 (11.8) |
| | | 0.4 | 79.0 (12.4) |
| | | 0.5 | 77.5 (13.1) |
| FedAvg | - | - | 10.0 (10.1) |
| TopK | - | - | 10.0 (10.1) |
| FedAvg + Trimmed-mean | batch size | 6 | 55.5 (14.5) |
| | | 8 | 56.3 (16.0) |
| | | 16 | 37.7 (15.6) |
| | local epochs | 2 | 41.0 (15.4) |
| | | 5 | 56.3 (16.0) |
| | | 10 | 21.0 (9.9) |
| | learning rate | 0.01 | 34.0 (15.5) |
| | | 0.05 | 38.3 (15.3) |
| | | 0.1 | 56.3 (16.0) |
| | | 0.15 | 10.0 (10.0) |
| | | 0.2 | 10.0 (10.0) |
| FedAvg + Multi-Krum | batch size | 6 | 19.0 (12.5) |
| | | 8 | 58.8 (15.8) |
| | | 16 | 36.7 (14.8) |
| | local epochs | 2 | 46.1 (15.9) |
| | | 5 | 58.8 (15.8) |
| | | 10 | 24.3 (11.7) |
| | learning rate | 0.01 | 15.3 (11.7) |
| | | 0.05 | 50.0 (16.2) |
| | | 0.1 | 58.8 (15.8) |
| | | 0.15 | 15.4 (11.9) |
| | | 0.2 | 10.0 (10.0) |
| SignSGD | batch size | 6 | 33.1 (15.6) |
| | | 8 | 39.7 (15.9) |
| | | 16 | 10.2 (10.1) |
| | local epochs | 2 | 10.2 (10.5) |
| | | 5 | 39.7 (15.9) |
| | | 10 | 41.5 (16.0) |
| | learning rate | 0.01 | 44.2 (15.8) |
| | | 0.05 | 41.9 (15.5) |
| | | 0.1 | 39.7 (15.9) |
| | | 0.15 | 35.8 (15.3) |
| | | 0.2 | 10.2 (10.1) |

successfully circumvents the target AGR. DYN-OPT is much stronger, because unlike STAT-OPT, it finds the largest $\gamma$ and uses a dataset tailored $\omega$.

## C    Missing Experiments

### C.1    FSL: Initialization Matters

Table 5: Comparing the performance of FSL with different random weight initialization algorithms with the performance of vanilla FedAvg for cross-device setting. Using Singed Kaiming Constant as weight initialization gives the best performance for all the datasets.

| Dataset | Metric | Algorithm | | | |
|---|---|---|---|---|---|
| | | FedAvg | FSL | | |
| | $W_{init} \sim$ | - | $\mathcal{X}_N$ | $\mathcal{N}_K$ | $U_K$ |
| MNIST LeNet N=1000 | Mean | 98.8 | 96.6 | 98.7 | 98.8 |
| | STD | 3.1 | 5.2 | 3.2 | 3.1 |
| | Min | 75.0 | 57.1 | 75.0 | 75.0 |
| | Max | 100 | 100 | 100 | 100 |
| CIFAR10 Conv8 N=1000 | Mean | 85.4 | 63.6 | 82.0 | 85.3 |
| | STD | 11.2 | 15.6 | 11.9 | 11.3 |
| | Min | 33.3 | 0 | 0 | 33.3 |
| | Max | 100 | 100 | 100 | 100 |
| FEMNIST LeNet N=3400 | Mean | 85.8 | 69.2 | 82.9 | 84.2 |
| | STD | 10.2 | 14.2 | 11.1 | 10.7 |
| | Min | 10.0 | 0 | 14.3 | 7.1 |
| | Max | 100 | 100 | 100 | 100 |

In FSL, the weight parameters are fixed throughout the FSL protocol and they are initialized randomly at the beginning of the protocol. It is very important to appropriately initialize the weights since the clients will find the subnetworks within these weights. We use three different distribution for initializing the weight parameters as follows:

**Glorot Normal** (Glorot & Bengio, 2010) where we denote by $\mathcal{X}_N$. Previous work Zhou et al. (2019) used this initialization to demonstrate that subnetworks of randomly weighted neural networks can achieve impressive performance.

**Kaiming Normal** (He et al., 2015) where we denote by $\mathcal{N}_k$ defined as $\mathcal{N}_K = \mathcal{N}\left(0, \sqrt{2/n_{\ell-1}}\right)$ where $\mathcal{N}$ shows normal distribution. $n_\ell$ shows the number of parameters in the $\ell$th layer.

**Singed Kaiming Constant** (Ramanujan et al., 2020) where all the wights are a constant $\sigma$ but they are assigned $\{+, -\}$ randomly. This constant, $\sigma$, is the standard deviation of Kaiming Normal. We show this initialization with $U_K$ as we are sampling from $\{-\sigma, +\sigma\}$ where $\sigma = \left(\sqrt{2/n_{\ell-1}}\right)$.

Table 5 shows the results of running FSL for three datasets under the three aforementioned initialization algorithms. We compare FSL with FedAvg and report the mean, standard deviation, minimum, and maximum of the accuracies for the clients' local subnetwork (for FSL) and local models (for FedAvg) at the end of FSL/FedAvg training. As we can see under three different random initialization, using Signed Kaiming Normal ($U_K$) results in better performance. We note from Table 5 that FSL with Signed Kaiming Normal ($U_K$) initialization achieves performance very close to the performance of FedAVg.

Note that, since the FSL clients update scores in each round, unlike initialization of weights, initialization of scores does not have significant impact on the final global subnetwork search. Therefore, we do not explore different randomized initialization algorithms for scores and simply use Kaiming Uniform initialization for scores.

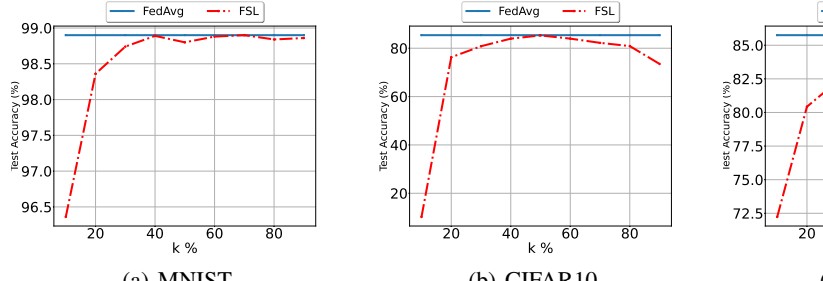

(a) MNIST             (b) CIFAR10             (c) FEMNIST

Figure 4: Comparing performance of FSL for different subnetwork sizes. $k$ (x-axis) shows the % of weights that each client is including in its subnetwork, test accuracy (y-axis) shows the mean of accuracies for all the clients on their test data. The chosen clients in each round send all the ranks to the server. FSL with subnetworks of $\in [40\%, 70\%]$ result in better performances.

Ramanujan et al. (2020) also considered these three initialization to find the best subnetwork in centralized machine learning setting. They also showed that using Singed Kaiming Normal gives the best supermasks. Our results align with their conclusions, hence we use Singed Kaiming Normal to initialize the weights and Kaiming Uniform to initialize the scores of global supernetwork.

## C.2   PERFORMANCES OF FSL WITH VARYING SIZES OF SUBNETWORKS

In FSL, each client uses Edge-Pop Algorithm (Ramanujan et al., 2020) and their local data to find a local subnetwork within a randomly initialized global network, which we call *supernetwork*. Edge-Pop algorithm use parameter $k$ which represents the % of all the edges in a supernetwork which will remain in the final subnetwork. For instance, $k = 50\%$ denotes that each client finds a subnetwork within a supernetwork that has half the number of edges as in the supernetwork.

Figure 4 illustrates how the performance of the global subnetwork in FSL varies with the size of subnetwork; note that, all of the clients collaborate to find the global subnetwork. We train nine FSL models with $k \in \{10, 20, 30, 40, 50, 60, 70, 80, 90\}\%$ and a FedAvg model (shown using a horizontal line); FedAvg model updates all the weights, hence it is a supermask with $k = 100\%$.

## C.3   PERFORMANCES OF FSL WITH DIFFERENT HETEROGENEOUS DATA DISTRIBUTION METHODS

Table 6 shows the performances of FSL and FedAvg using different methods of non-iid assignment. We distribute the data between 1000 clients with two methods: (I) Dirichlet distribution with $\beta = 1$ similar to (Reddi et al., 2020; Hsu et al., 2019) and (II) each client has data of 2 random classes similar to (McMahan et al., 2017). In this table, we can see that FSL can achieve the same performance of FedAvg in different heterogeneous data distributions.

## D   MISSING DETAILS OF EDGE-POPUP AND FSL ALGORITHM

Suppose in a fully connected neural network, there are $L$ layers and layer $\ell \in [1, L]$ has $n_\ell$ neurons, denoted by $V^\ell = \{V_1^\ell, ..., V_{n_\ell}^\ell\}$. If $I_v$ and $Z_v$ denote the input and output for neuron $v$ respectively, then the input of the node $v$ is the weighted sum of all nodes in previous layer, i.e., $I_v = \sum_{u \in V^{\ell-1}} W_{uv} Z_u$. Here, $W_{uv}$ is the weight of the edge connecting $u$ to $v$. Edge-popup algorithm tries to find subnetwork $E$, so the input for neuron $v$ would be: $I_v = \sum_{(u,v) \in E} W_{uv} Z_u$.

**Updating scores.** Consider an edge $E_{uv}$ that connects two neurons $u$ and $v$, $W_{uv}$ be the weight of $E_{uv}$, and $s_{uv}$ be the score assigned to the edge $E_{uv}$ by Edge-popup algorithm. Then the edge-popup algorithm removes edge $E_{uv}$ from the supermask if its score $s_{uv}$ is not high enough. Each iteration of supermask training updates the scores of all edges such that, if having an edge $E_{uv}$ in subnetwork reduces loss (e.g., cross-entropy loss) over training data, the score $s_{uv}$ increases.

Table 6: Comparing the performance of FSL and FedAvg for cross-device setting using two methods of data assignment. We distribute the data between 1000 clients with two methods: (I) Dirichlet distribution with $\beta = 1$ and (II) each client has data of 2 random classes.

| Dataset | Type of Non-IID | Metric | Algorithm | |
|---|---|---|---|---|
| | | | FedAvg | FSL |
| MNIST LeNet N=1000 | Dirichlet Distribution $\beta = 1$ | Mean | 98.8 | 98.8 |
| | | STD | 3.1 | 3.1 |
| | | Min | 75.0 | 75.0 |
| | | Max | 100 | 100 |
| | Randomly 2 classes assigned to each client | Mean | 98.4 | 98.3 |
| | | STD | 4.3 | 4.1 |
| | | Min | 70.0 | 80.0 |
| | | Max | 100 | 100 |
| CIFAR10 Conv8 N=1000 | Dirichlet Distribution $\beta = 1$ | Mean | 85.4 | 85.3 |
| | | STD | 11.2 | 11.3 |
| | | Min | 33.3 | 33.3 |
| | | Max | 100 | 100 |
| | Randomly 2 classes assigned to each client | Mean | 70.6 | 70.9 |
| | | STD | 21.9 | 19.2 |
| | | Min | 0 | 10.0 |
| | | Max | 100 | 100 |

The algorithm selects top k% edges (i.e., finds a subnetwork with sparsity of k%) with highest scores, so $I_v$ reduces to $I_v = \sum_{u \in V^{\ell-1}} W_{uv} Z_u h(s_{uv})$ where $h(.)$ returns 1 if the edge exists in top-k% highest score edges and 0 otherwise. Because of existence of $h(.)$, which is not differentiable, it is impossible to compute the gradient of loss with respect to $s_{uv}$. Recall that, the edge-popup algorithm use straight-through gradient estimator (Bengio et al., 2013) to compute gradients. In this approach, $h(.)$ will be treated as the identity in the backward pass meaning that the upstream gradient (i.e., $\frac{\partial L}{\partial I_v}$) goes straight-through $h()$. Now using chain rule, we can derive $\frac{\partial L}{\partial I_v} \frac{\partial I_v}{\partial s_{uv}} = \frac{\partial L}{\partial I_v} W_{uv} Z_u$ where $L$ is the loss to minimize. Then we can SGD with step size $\eta$ to update scores as $s_{uv} \leftarrow s_{uv} - \eta \frac{\partial L}{\partial I_v} Z_u W_{uv}$.

Ramanujan et al. (2020) proved that when edge $(a, b)$ replaces $(c, b)$ in layer $\ell$ and the rest of the subnetwork remains fixed then the loss of the supermask learning decreases (provided the loss is sufficiently smooth). Motivated by their proof, we can show when these two edges are swapped in FSL, the loss decreases for FSL optimization too.

**Theorem 1:** when edge $(a, b)$ replaces $(c, b)$ in layer $\ell$ and the rest of the subnetwork remains fixed then the loss of the FSL optimization will decrease (provided the loss is sufficiently smooth).

*proof.* First, we know that the optimization problem of FSL is as follow:

$$\min_{R_g} F(\theta^w, R_g) = \min_{R_g} \sum_{i=1}^{N} \lambda_i L_i(\theta^w \odot \mathbf{m}) \quad s.t. \tag{5}$$

$$\mathbf{m}[R_g < t] = 0 \quad \text{and} \quad \mathbf{m}[R_g >= t] = 1 \tag{6}$$

where $\lambda_i$ shows the importance of the $i^{th}$ client in empirical risk minimization which $\lambda_i = \frac{1}{N}$ gives same importance to all the participating clients. $\mathbf{m}$ is the final mask that contains the edges of top ranks, and $L_i$ is the loss function for the $i$th client. $\theta^w \odot \mathbf{m}$ shows the subnetwork inside the random $\theta^w$ that all clients unanimously vote for. In this optimization, the FSL clients try to minimize $F$ by finding the best global ranking $R_g$.

We now wish to show $F(\theta^w, R_g^{t+1}) < F(\theta^w, R_g^t)$ when in FSL round $t+1$, the edge $(a, b)$ replaces $(c, b)$ in layer $\ell$ and the rest of the subnetwork remains fixed. Suppose global rank of edge $(a, b)$ was $R_g^t[(a, b)]$ and global rank of edge $(c, b)$ was $R_g^t[(c, b)]$ in round $t$, so we have:

$$R_g^t[(a,b)] < R_g^t[(c,b)] \tag{7}$$

$$R_g^{t+1}[(a,b)] > R_g^{t+1}[(c,b)] \tag{8}$$

where the order of all remaining global ranks remain fixed, and only these two edges are swapped in global ranking. Now let $s_{ab}^{t,i}$ shows the score of weight $w_{ab}$ in round $t$ and client $i^{th}$ and $s_{ab}^{t+1,i}$ shows the updated score of it after local training. As in our majority vote, we are calculating the sum of the reputation of edges we will have:

$$\sum_{i=1}^{N} s_{ab}^{t,i} < \sum_{i=1}^{N} s_{cb}^{t,i} \tag{9}$$

$$\sum_{i=1}^{N} s_{ab}^{t+1,i} > \sum_{i=1}^{N} s_{cb}^{t+1,i} \tag{10}$$

We also know that Edge-popup algorithm updates the scores in the $i^{th}$ client as follow:

$$s_{ab}^{t+1,i} = s_{ab}^{t,i} - \eta \frac{\partial L}{\partial I_a} Z_a W_{ab} \tag{11}$$

Based on 9, 10 and 11, we can say:

$$\sum_{i=1}^{N} s_{ab}^{t,i} - \sum_{i=1}^{N} s_{cb}^{t,i} < \sum_{i=1}^{N} s_{ab}^{t+1,i} - \sum_{i=1}^{N} s_{cb}^{t+1,i} \tag{12}$$

We also know that:

$$\sum_{i=1}^{N} \left( s_{ab}^{t+1,i} - s_{ab}^{t,i} \right) = \sum_{i=1}^{N} \left( -\eta \frac{\partial L^i}{\partial I_a^i} Z_a^i W_{ab} \right) \tag{13}$$

$$\sum_{i=1}^{N} \left( s_{cb}^{t+1,i} - s_{cb}^{t,i} \right) = \sum_{i=1}^{N} \left( -\eta \frac{\partial L^i}{\partial I_c^i} Z_c^i W_{cb} \right) \tag{14}$$

Based on 12, 13 and 14, we can say:

$$\sum_{i=1}^{N} \left( \frac{\partial L^i}{\partial I_c^i} Z_c^i W_{cb} \right) > \sum_{i=1}^{N} \left( \frac{\partial L^i}{\partial I_a^i} Z_a^i W_{ab} \right) \tag{15}$$

So based on 15, and what Ramanujan et al. (2020) proved for each supermask training we can show 16. We assume that loss is smooth and the input to the nodes that their edges are swapped are close before and after the swap.

$$\sum_{i=1}^{N} \left( L_i(\theta^w \odot \mathbf{m^{t+1}}) \right) < \sum_{i=1}^{N} \left( L_i(\theta^w \odot \mathbf{m^t}) \right) \tag{16}$$

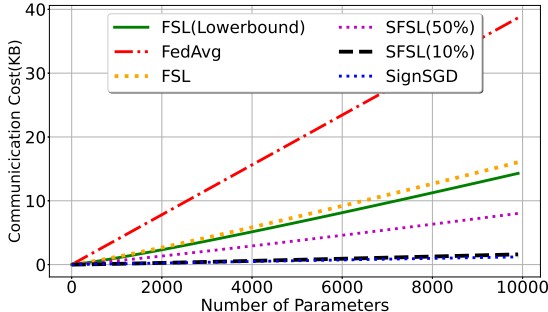

Figure 5: Communication cost Analysis. Please note that the download communication cost of all SFSLs would be the same as FSL.

that means:

$$F(\theta^w, R_g^{t+1}) < F(\theta^w, R_g^t) \tag{17}$$

## E    MISSING DETAILS ABOUT COMMUNICATION COST COMPARISON

One of the features of the FSL training is its communication efficiency. In Section 5.2, we show that if the FSL clients send and receive rankings, the communication cost will be $\sum_{\ell \in [L]} n_\ell \times \log(n_\ell)$ bits per client. In this section, we are providing a lower bound on the FSL communication cost, and then compare it with FedAvg and SignSGD.

**Lowerbound of communication cost of FSL:** Since the FSL clients send and receive layer-wise rankings of indices, i.e., integers $\in [0, n_\ell - 1]$, for layer $\ell$, there are $n_\ell!$ possible permutations for layer $\ell \in [L]$. If we use the best possible compression method in FSL, an FSL client needs to send and receive $\sum_{\ell \in [L]} \log(n_\ell!)$ bits. Therefore, the download and upload bandwidth for each FSL client would be $\sum_{\ell in [L]} \log(n_\ell * (n_\ell - 1) * ... * 2 * 1) = \sum_{\ell \in [L]} \sum_{i=1}^{n_\ell} log(i)$ bits. Please note that in our experiment, FSL clients send and receive the rankings without any further compression, and $\sum_{\ell \in [L]} \sum_{i=1}^{n_\ell} log(i)$ just shows a lower-bound of communication cost of FSL. In Section 6.1, we measure the performance and communication cost of FSL with other existing FL compressors signSGD (Bernstein et al., 2019) and TopK (Aji & Heafield, 2017; Alistarh et al., 2018a). In Figure 5, we compare the communication cost of one client per FL round for FedAvg, SignSGD, and different variant of FSL for different number of parameters.

Similar work in this domain is LotteryFL (Li et al., 2020a), a personalization framework that each FL client learns a lottery ticket network (LTN) by pruning the base model using Lottery Ticket hypothesis (Frankle & Carbin, 2019). In LotteryFL, each client sends and receives the update for its subnetwork, and at the end, they have an extra step for personalization. FSL is different from LotteryFL as the FSL clients find subnetworks within a random and fixed network and send the ranks of their subnetwork edges instead of what LotteryFL clients do that train their weights and find a subnetwork by freezing some weights and send their actual model update. LotteryFL is based on FedAvg that the clients can send any update to the server, which is vulnerable to the same attacks that existed for FedAVG. In terms of communication cost, FSL is very close to LotteryFL as they report 1.81x improvement over CIFAR10 which is close to FSL and SFSL(50%) which provide 1.53x, 3.09x improvement respectively over CIFAR10.

## F    ADDITIONAL COMPARISONS

Figure 6 is showing the learning curve of FSL for different numbers of local epochs for CIFAR10 experiment. On the x-ais we have accumulated communication cost: $e \times 13.1 \times 2 \times 25$ MB where $e$ is the FSL round, 13.1MB is the cost of FSL per client, 2 is for download+upload cost, and 25 clients are selected in each round.

Table 7: The effect of other settings on performance of FSL trained on CIFAR10 distributed over 1000 clients using Dirichlet distribution. The **bold** shows the value we used in our experiments.

| Method | hyperparameter | value | Test Accuracy with 10% malicious |
|---|---|---|---|
| FSL | Number of participants (n) | 15 | 84.8 (11.3) |
| | | **25** | **85.3 (11.3)** |
| | | 50 | 84.9 (11.2) |
| | local epochs (E) | 2 | 82.2 (12.0) |
| | | **5** | **85.3 (11.3)** |
| | | 10 | 83.5 (11.9) |
| | Non-iid degree ($\beta$) | **1** | **85.3 (11.3)** |
| | | 10 | 85.6 (11.1) |
| | | 100 | 85.6 (10.9) |

(a) CIFAR10 (Test Accuracy)

(b) CIFAR10 (Test Loss)

Figure 6: Comparing performance of FSL for different local epochs.

Table 7 is showing the effect of other settings on performance of FSL trained on CIFAR10 distributed over 1000 clients using Dirichlet distribution. The **bold** shows the value we used in our experiments.

