# OpenReview forum: "FSL: Federated Supermask Learning"
_ICLR.cc/2022/Conference — ICLR 2022 Submitted_

### Official Review · Reviewer_JJjz · 2021-10-28

**Correctness:** 3
**Technical Novelty And Significance:** 2
**Empirical Novelty And Significance:** Not applicable
**Recommendation:** 6
**Confidence:** 4

**Main Review:**

I enjoyed reading this paper. The authors accompany their explanation with an illustrative example which helps understanding.
Regardless, several things remain unclear.

High-level: Throughout the text the authors refer to robustness and communication as 'the' important issues in FL. I would argue that they are two important issues, but there are also many other important issues, such as differential privacy, efficiency, data/hardware/network heterogeneity, questions of fairness and representation among many others.

Communication:
- I am confused by the authors calculation of the communication requirements of SFL (Appendix E).
In order to index all possible permutations of a layer with $n_l$ weights, we need $\log_2(n_l!)$ bits. the computation of that number does not require Stirling's approximation, since $\log(n_l!) = \log(n_l*(n_l -1) * ( n_l - 2)* ... * 2 * 1) = \sum_1^{n_l} \log(i)$.
Using the provided numbers in Table 3 for the Cifar10 model, I compute
sum([np.ceil(np.sum(np.log2(np.arange(2,l+1))))/(8 * 1024 * 1024) for l in layers]) = 11.698538184165955mb for a single message, instead of the 13.1MB that the authors claim.
- Unfortunately the authors ignore a mayor practical aspect of this approach. What is the computational overhead in finding the appropriate index for a given permutation? I landed on the Lehmer code (https://en.wikipedia.org/wiki/Lehmer_code#Encoding_and_decoding) for computing the index, which seems to scale quadratically in $n_l$. Using https://gist.github.com/lukmdo/7049748 for a quick test, encoding a random permutation of length $2359296$ takes approx 42minutes on a modest Desktop-CPU. I would encourage the authors to discuss this in light of the cross-device setting and resource-constrained devices.
The alternative approach of encoding a given permutation as list of integers would require $n_l*\log(n_l)$ bits, which amounts to
sum([(l*np.ceil(np.log2(l)))/(8*1024*1024) for l in layers]) = 13.069404602050781mb. Now that I compute that number I realise that's the budget the authors claim, i.e. they seem to not index the permutation. I encourage the authors to clarify.

Robustness to Attacks:
- The authors claim that worst-case attack on FSL requires inverting the order of rankings. I would argue that a stronger attack might be considered when considering potential effects across all layers of the network. I.e. can a malicious client cause more damage by coordinating across layers?
- Algorithm 3 in Appendix A.1 seems to suggest that Voting in Line 11 happens only across malicious clients. This goes in line with Argument 3)in the main-text concerning collusion. Why is this form of collusion the strongest form of attack? What are the consequences of each client independently reversing their rankings? On the other hand, can there be a stronger attack by not performing Vote, followed by reversing the order - and instead some other finding of consensus between malicious clients to maximally attack the model's performance?
- Can you validate your theoretical curves in Figure 2 by measuring flips in your empirical experiments?

Sparse-FSL:
The authors discuss that client's don't send the bottom % of their rankings and the server assumes a reputation of 0 for those entries. In the end of section 6.1 the authors discuss that additionally, download-costs are reduced. Information about what the client's assume for the left-out entries is missing, however.
In the same paragraph, the authors argue that with 50% ranks sent, the communication is cut in half. Relating to above discussion about communication costs, there needs to be an encoding scheme behind this sparsification. I.e. if for $p\%$ remaining rankings, you'd need $p n_l\log_2(p n_l)$ for communicating the reduced rankings, plus a binary mask of $n_l$ bits.

Experiments:
- The authors state that they give 'mean of all client's test-accuracies'. Are those on fine-tuned local models? Are those weighted by local client's data-set sizes? Alternatively, keep the test-set on the server and evaluate the test-set on the server. Afaik this is the standard practice.
- I examined the CIFAR10 notebook provided as supplementary. The way the authors split the data cross clients leads to some data-points not being used at all apparently, i.e. te_count + tr_count = 59886
- Please provide your hyper parameter selection strategy: Did you optimize hyper parameters across baselines and FSL independently? Which ranges did you consider?
- Based on how many seeds did you report the mean+std (Table 1 and 2). The deviations seem to be very high, i.e. for many experiments, the averages lie within the std of each other. It might be better to report the standard-error and repeat experiments more often to get a more reliable estimate. Additionally, it might also be a good idea to average evaluation-accuracies across the last few epochs to get a more robust estimate. Please plot stderror on below mentioned learning curves accordingly


Baselines:
- For SignSGD, do you communicate the bit-mask ever local gradient-step as originally proposed - or do you communicate a bit-mask after ever local Epoch E, as would be in line with Federated Learning? SignSGD cannot really be assumed to be a proper Federated algorithm due to the frequent communication - indeed it was proposed for data-centre applications.
In case these stated Upload / Download costs are not happening at the same frequencies as for e.g. FedAvg of FSL, Table 1 is misleading.
- Please plot learning curves where on the x-axis we have accumulated communication-budget and on the y-axis we see validation accuracy.
- Apart from binarising gradients, there is also the option to quantise gradients to b bits. A basic baseline for update expression is to do group-wise (e.g. per-layer) quantization to a b-bit grid that uniformly divides [min(x),max(x)] for x being the group of parameter-updates. The min/max values of the grid need to be communicated to the server. Stochastic quantization is important here.
Further compression can be achieved by performing vector-quantisation. A relatively recent work in this domain is HSQ (https://arxiv.org/abs/1911.04655), which targets the FL setting explicitly.


Some minor aspects:
- In Algorithm 3 you index with $mu$, as well as only $u$ and only $m$. What is the meaning of these individual or joint indices?
- In Appendix B.2 and the main text, you should describe your sampling approach better, i.e. you never mention that you assign *labels* based on the Dirichlet samples. A convenient source to cite (as well as maybe improve your implementation with) is https://arxiv.org/abs/2003.00295.
- Why are the authors citing Minka(2000) when mentioning the Dirichlet distribution? The cited work discussed parameter estimation for the Dirichlet distribution, which is not on-topic here.
- page 14 bottom, 'sparsification' no capitalisation
-  Introduction First paragraph: How is 'Google' an example of a server?
- When describing a genderless object, such as a client, one should use 'their' instead of 'his' or 'her'. E.g. Abstract 3 introduction: ... send their local edge ranking...; Also: 3.1: '... on their data...'

Final disclaimer: I am not an expert in defences and attacks in the FL setup and will defer to other reviewers in their evaluation

**Summary Of The Paper:**

The authors show how to apply the method of Ramanujan et al. (2020) and Wortsman et al. (2020) to the federated setting. They show that the proposed method, FSL, has desirable properties in reducing communication overhead and improving robustness to malicious attacks on the resulting model compared to other approaches.


**Summary Of The Review:**

In Summary, the authors apply an existing algorithm to the FL setup, including a novel voting-algorithm that applies to this algorithm. The execution of the paper has some flaws that I would like to see corrected before the paper is ready for publication.

---

> ### Author Response · Authors · 2021-11-17
> **Authors' Response to Reviewer JJjz (part 1)**
>
> Thank you for your helpful feedback. We would like to address your concerns below.
>
> ***
> **High-level:**
> - **C1:** [Re: "Throughout the text the authors refer to robustness and communication as 'the' important issues in FL. I would argue that they are two important issues, but there are also many other important issues, such as differential privacy, efficiency, data/hardware/network heterogeneity, questions of fairness and representation among many others."]
>
> **Ans1:** We agree that there are other important issues facing FL, but our focus in this work is the two issues of robustness and communication efficiency. We will adjust our writing to make this clarified.
>
> ***
> **Communication:**
> - **C2:** [Re: "Communication Cost"]
>
> **Ans2:** In all of our experiments (including table 3), the local and global rankings are sent and received without any further compression. Therefore the size of each ranking is $\sum_{\ell \in [L]} n_{\ell} * log(n_{\ell})$ (e.g., 13.1MB for the CIFAR10 dataset). However, if we utilize a ranking compression, the lowerbound on communication cost would be the same as the reviewer mentioned $\sum_{\ell \in [L]} \log(n_{\ell}!)=\sum_{\ell \in [L]}  \log\left(n_{\ell} * (n_{\ell}-1) * ... *2 *1\right)=\sum_{\ell \in [L]} \sum_{i=1}^{n_{\ell}} \log(i)$. We just provided this as a lowerbound of communication cost for FSL, but in all experiments, we use simple ranking transmission. We have clarified this in section E.
>
> ***
> - **C3:** [Re: "Computation of calculating the ranks" ]
>
> **Ans3:** We did not compute permutations in FSL. We just provided the $\sum_{\ell in [L]} \log(n_{\ell}!)$ as a lowerbound of communication cost if further compression is used. FSL clients send and receive the ranks without calculating any permutation, they just train scores and find argsort(scores) as their ranks.
> We use following function to calculate the rankings of a  vector of scores:\
> **def function** Findrank(scores):\
> &nbsp;&nbsp;    out = scores.detach().clone();\
> &nbsp;&nbsp;    _, idx = scores.flatten().sort();\
> &nbsp;&nbsp;    **return** idx
>
> This function involves a simple pytorch sort, for example in experiments of CIFAR10, each client converts Conv8() scores into ranks in n 0.0232 seconds on a modest Desktop-CPU. Table 3 shows the number of parameters in each layer.
> We have added more text to Section E to emphasize that the permutation part is just part of the lowerbound of FSL communication cost.
>
> ***
> **Robustness to Attacks:**
> - **C4:** [Re: "The authors claim that the worst-case attack on FSL requires inverting the order of rankings. I would argue that a stronger attack might be considered when considering potential effects across all layers of the network. I.e. can a malicious client cause more damage by coordinating across layers"]
>
> **Ans4:**
> In our attack, the malicious client inverts the ranking of each layer separately. For example, in our experiments of MNIST, each network has 4 layers (Table 3), so the malicious client will invert the ranking of each layer separately. We have clarified this in Section 5.1.  We justify that this is the strongest attack since FSL server gives reputation to each edge based on its position in the ranking, and it sums all of the reputations. Therefore, inverting the rankings causes the reputation of good edges to come down since the malicious client is injecting low reputations for them and bad edges are coming up in global ranking by increasing their reputation.
> SingSGD[a] proved that the strongest attack on their system is to invert the sign of their updates too. FSL could be a generalization of SignSGD that inverting rankings is the strongest attack.
>
> [a]: Jeremy Bernstein et al.  "signsgd with majority vote is communication efficient and fault-tolerant", ICLR, 2019

---

> > ### Author Response · Authors · 2021-11-17
> > **Authors' Response to Reviewer JJjz (part 2)**
> >
> > - **C5:** [Re: "Algorithm 3 in Appendix A.1 seems to suggest that Voting in Line 11 happens only across malicious clients. This goes in line with Argument 3)in the main-text concerning collusion. Why is this form of collusion the strongest form of attack? What are the consequences of each client independently reversing their rankings? On the other hand, can there be a stronger attack by not performing Vote, followed by reversing the order - and instead some other finding of consensus between malicious clients to maximally attack the model's performance"]
> >
> > **Ans5:** We follow the standard assumption from the literature that the attackers can collude to improve their attack’s impact. Such an attacker has more knowledge about the true global ranking as it wants to invert the best available global ranking in a given round, i.e. putting bad edges in high ranks and good edges in low ranks. Malicious clients collude to have a better understanding of the global ranking since each client has access to a small number of data points. For example, in our CIFAR10 experiments, some clients have only 30 images, they cannot find a good estimation of global ranking on their own.
> >
> > Note that, the strongest attacker is the one with the knowledge of the rankings of all the malicious as well as benign clients. However, this is highly impractical assumption from a practical point of view, hence we omit it.
> >
> > ***
> > - **C6:** [Re: "Can you validate your theoretical curves in Figure 2 by measuring flips in your empirical experiments"]
> >
> > **Ans6:** We have added Figure 6 (please check the revised version of our submission) showing the number of votes for a particular edge in the first layer of LeNet (trained on MNIST). We also showed the thresholds for each malicious ratio that if the number of votes is less than them, the adversary can change the decision about this edge.
> >
> > ***
> > **SFSL:**
> > - **C7:** [Re: "Sparse-FSL: The authors discuss that client's don't send the bottom % of their rankings and the server assumes a reputation of 0 for those entries. In the end of section 6.1 the authors discuss that additionally, download-costs are reduced. Information about what the client's assume for the left-out entries is missing, however. In the same paragraph, the authors argue that with 50% ranks sent, the communication is cut in half."]
> >
> > **Ans7:** We have clarified in Section 6.1 that in SFSL, the clients only send %x top ranks, and the server assumes zero for missing ranks. This makes the upload bandwidth half for SFSL 50% or one-tenth for SFSL 10%. However, the download bandwidth would be the same as the clients do not have any assumption about the missing ranks. Also in Table 1, the download consumption of FSL, SFSL Top 50% and SFSL Top 10% were the same as the server sends all the global rankings to the clients.
> >
> > ***
> > **Experiments:**
> > - **C8:** [Re: "The authors state that they give 'mean of all client's test-accuracies'. Are those on fine-tuned local models? Are those weighted by local client's data-set sizes? Alternatively, keep the test-set on the server and evaluate the test-set on the server. Afaik this is the standard practice."]
> >
> > **Ans8:** We follow the evaluation strategy of [a,b] and at the end of training for each FL algorithm, we calculate the test accuracies of each client and we report the mean and standard deviation of all the test accuracies. No, we do not do any fine-tuning at the end of training for any of the algorithms. No, they are not weighted by local-data sizes. We report a simple average as $\frac{1}{N} \sum_{i=1}^N \text{test}(\text{globalmodel}, D_i^{\text{test}})$ without considering the data sizes that each client holds.
> > Calculating the mean of accuracies for all the clients is the same as keeping all the test data in one place and calculating the accuracy on it.
> >
> > [a]: Yu et al. "Salvaging federated learning by local adaptation." arXiv preprint arXiv:2002.04758 (2020).\
> > [b]: Li et al. "Ditto: Fair and robust federated learning through personalization." ICML 2021.
> >
> > ***
> > - **C9:** [Re: I examined the CIFAR10 notebook provided as supplementary. The way the authors split the data cross clients leads to some data-points not being used at all apparently, i.e. te_count + tr_count = 59886]
> >
> > **Ans9:** Due to Dirichlet distributed datasets, we are missing 114 data points out of 6000 images in CIFAR10. All the previous works [a,b] which use such Dirichlet distributed local datasets are bound to miss some points. Note that this does not change our core empirical analyses.
> >
> > [a]: Reddi et al. "Adaptive federated optimization." arXiv preprint arXiv:2003.00295 (2020).
> > [b]: Shejwalkar et al. "Manipulating the Byzantine: Optimizing Model Poisoning Attacks and Defenses for Federated Learning", NDSS21.

---

> > > ### Author Response · Authors · 2021-11-17
> > > **Authors' Response to Reviewer JJjz (part 3)**
> > >
> > > - **C10:** [Re: "Please provide your hyperparameter selection strategy: Did you optimize hyper parameters across baselines and FSL independently? Which ranges did you consider"]
> > >
> > > **Ans10:** Yes, we optimize the hyperparameters based on FSL and FedAvg independently. We have added Section B.2 and Table 4 to explain more about our hyperparameter tuning.
> > >
> > > ***
> > > - **C11**: [Re: "Based on how many seeds did you report the mean+std (Table 1 and 2). The deviations seem to be very high, i.e. for many experiments, the averages lie within the std of each other. It might be better to report the standard-error and repeat experiments more often to get a more reliable estimate. Additionally, it might also be a good idea to average evaluation-accuracies across the last few epochs to get a more robust estimate. Please plot stderror on below mentioned learning curves accordingly"]
> > >
> > > **Ans11:** Please note that the STD in Tables 1 and 2 shows the standard deviation of the performance of the final global model across different clients. This is a standard metric [a,b] that is more appropriate to consider as it measures the average performance as well as fairness of a given FL algorithm.
> > > As [a,b] defined, a more fair model is a model that produces more uniform results across different clients. It means that model M1 is more fair than model M2 if the STD[test(M1, i) for i in [N]] < STD[test(M1, i) for i in [N]].
> > >
> > > [a]: Li, Tian et al. "Ditto: Fair and robust federated learning through personalization." ICML 2021.\
> > > [b]: Smith, Virginia, et al. "Federated Multi-Task Learning." NIPS. 2017.
> > >
> > > ***
> > > **Baselines:**
> > > - **C12:** [Re: "For SignSGD, do you communicate the bit-mask ever local gradient-step as originally proposed - or do you communicate a bit-mask after ever local Epoch E, as would be in line with Federated Learning? SignSGD cannot really be assumed to be a proper Federated algorithm due to the frequent communication - indeed it was proposed for data-centre applications. In case these stated Upload / Download costs are not happening at the same frequencies as for e.g. FedAvg of FSL, Table 1 is misleading"]
> > >
> > > **Ans12:** Yes, SignSGD was proposed for FedSGD where bit-masks are communicated after every gradient-step, but due to its efficacy in FedAvg, it is also used in the FL setting where bit-masks are communicated after E local epochs.  The upload/download in Table 1 happen at the same frequencies, i.e., after E local epochs. We have reported the performance of SignSGD for different numbers of local epochs={2,5,10} in Table 4.
> > >
> > > ***
> > > - **C13:** [Re: "Please plot learning curves where on the x-axis we have accumulated communication-budget and on the y-axis we see validation accuracy."]
> > >
> > > **Ans13:** We have added Figure 5 showing the learning curve of FSL for different numbers of local epochs. On the x-axis, we have accumulated communication cost: $e \times 13.1 \times 2 \times 25$ MB where $e$ is the FSL round, 13.1MB is the cost of FSL per client, 2 is for download+upload cost, and 25 clients are selected in each round.
> > >
> > > ***
> > > - **C14:** [Re: "Apart from binarising gradients, there is also the option to quantise gradients to b bits. A basic baseline for update expression is to do group-wise (e.g. per-layer) quantization to a b-bit grid that uniformly divides [min(x),max(x)] for x being the group of parameter-updates. The min/max values of the grid need to be communicated to the server. Stochastic quantization is important here. Further compression can be achieved by performing vector-quantisation."]
> > >
> > > **Ans14:** Please note that, as we mentioned in the Introduction, a major part of our motivation is to design a robust FL algorithm that is also communication efficient. There are numerous techniques to compress the FL updates. But, they require their clients to train weights and send compressed weights to the server to perform simple averaging, and hence, are not robust to poisoning attacks. SignSGD is shown to be robust to poisoning attacks [a] hence we use it in our evaluations.
> > >
> > > [a]: Jeremy Bernstein et al.  "signsgd with majority vote is communication efficient and fault tolerant".  ICLR, 2019
> > >
> > > ****
> > > **Some minor aspects**
> > > - **C15:** [Re: "In Algorithm 3 you index with mu, as well as only u and only m. What is the meaning of these individual or joint indices?"]
> > >
> > > **Ans15:** We tried to show the normal users with index u, malicious users with mu, and something related to all of the malicious as m (like R_m^t: the global ranking that malicious clients find in round t)
> > >
> > > ***
> > >
> > > - **C16:** [rest of minor comments]:
> > >
> > > **Ans16:** We have fixed the minor comments that the reviewer mentioned.

---

> > > > ### Comment · Reviewer_JJjz · 2021-11-23
> > > > **Thank you for the rebuttal**
> > > >
> > > > I thank the authors for addressing all my comments.
> > > >
> > > > I have a few open points that I am curious about:
> > > > C4/C5: You are reiterating that the per-layer inverting of ranks constitutes the strongest possible attack. I see a connection but not a correspondence to signsgd. There, each individual entry is 'inverted' by flipping the sign, but not a layer-wise ranking.
> > > > I am imagining a situation in which all malicious clients come together, analyse their collective local datasets and design an update that maximally attacks the next-round server-side model. It seems to me that the per-layer, per-client perspective is not the strongest possible attack. E.g. could it be in the attachers' interests to leave one layer's ranking mostly benign because the effect of disturbing another layer's ranking is much more detrimental that way?
> > > > In case I am wrong about this scenario, can you provide a formal prove why the layer-wise per-client inversion is the "strongest poisoning attach", as you state in the paper?
> > > >
> > > > C6: You mean Figure 3?
> > > > C11: I see, you are providing std across clients. I am assuming you are not repeating your individual experiments for several seeds then? This is not good practice and I encourage you to strengthen your empirical analysis by running more seeds.
> > > > C13: You provided learning curves for SFL only. The only insight from Figure 6 is that E=5 seems to be best for that dataset. What I wanted to see as part of comparisons with baselines are learning curves for the methods in Table 1.
> > > > C14: A trivial baseline to make scalar quantization robust to large-magnitude weight-updates is to restrict the grid-range for decoding of updates. The server could define an upper/lower bound for the grid-range by choosing a (EMA) quantile of reported grid-ranges.
> > > >
> > > > Since the authors have addressed most of my questions, I will raise my score to 6.

---

> > > > > ### Author Response · Authors · 2021-11-29
> > > > > **Authors' Response to Reviewer JJjz**
> > > > >
> > > > > Thank you for your helpful feedback. We would like to address your concerns below.
> > > > >
> > > > > -----
> > > > >
> > > > > - **CC1:** "(C4/C5) You are reiterating that the per-layer inverting of ranks constitutes the strongest possible attack...."
> > > > >
> > > > > **AA1:** In our experiments, we used the same approach that you mentioned as the strongest form of attack where all the malicious clients collude, find their benign rankings, combine them, and finally invert the final vote. However, in our theoretical analysis, we used a simpler assumption (similar to what SignSGD assumes in their proofs) as each malicious client is inverting its own benign ranking without any collaboration with other malicious clients. We used this assumption as we assume their votes have a binomial distribution (section A.2) for showing our upper-bound on the failure probability of the simple vote mechanism.
> > > > >
> > > > > ------
> > > > >
> > > > > - **CC2:** "(C6) You mean Figure 3?"
> > > > >
> > > > > **AA2:** Yes, Figure 3 is showing the number of votes for a particular edge in the first layer of LeNet (trained on MNIST).
> > > > >
> > > > > ------
> > > > >
> > > > > - **CC3:** "(C11) I see, you are providing std across clients. I am assuming you are not repeating your individual experiments for several seeds then? This is not good practice and I encourage you to strengthen your empirical analysis by running more seeds."
> > > > >
> > > > > **AA3:** We appreciate your comment and we will do that in the final version. Although the result of FSL is stable as we run enough experiments.
> > > > >
> > > > > ----
> > > > >
> > > > > - **CC4:** "(C13) You provided learning curves for SFL only. The only insight from Figure 6 is that E=5 seems to be best for that dataset. What I wanted to see as part of comparisons with baselines are learning curves for the methods in Table 1."
> > > > >
> > > > > **AA4:** We will provide the learning curve for other baselines in the final version as the draft is locked now.
> > > > >
> > > > > -------
> > > > >
> > > > > - **CC5:** "(C14): A trivial baseline to make scalar quantization robust to large-magnitude weight-updates is to restrict the grid-range for decoding of updates. The server could define an upper/lower bound for the grid-range by choosing a (EMA) quantile of reported grid-ranges."
> > > > >
> > > > > **AA5:** We believe we have considered Multi-Krum and Trimmed-Mean as the two state-of-the-art Byzantine-robust aggregation mechanisms. They are working similar to your approach as follow:
> > > > >
> > > > > - Multi-Krum selects an update using Krum and adds it to a selection set, S. Krum selects the gradient from the set of its input gradients that is closest to its n − m − 2 neighboring gradients in the squared Euclidean norm space. Multi-Krum repeats this for the remaining updates (which remain after removing the update that Krum selects) until S has c updates such that n − c > 2m + 2, where n is the number of selected clients and m is the number of compromised clients in a given round. Finally, Multi-Krum averages the updates in S.
> > > > >
> > > > > - Trimmed-mean aggregates each dimension of input updates separately. It sorts the values of the jth-dimension of all updates. Then it removes m (i.e., the number of compromised clients) of the largest and smallest values of that dimension, and computes the average of the rest of the values as its aggregate of the dimension j.

---

### Official Review · Reviewer_XrNo · 2021-10-31

**Correctness:** 3
**Technical Novelty And Significance:** 3
**Empirical Novelty And Significance:** 2
**Recommendation:** 6
**Confidence:** 4

**Main Review:**

# Strengths
* The idea of introducing supermask learning (on a randomly initialized network) for federated learning is interesting.
* The algorithm leverages the ideas like edge ranking order and majority voting to achieve the reduced communication cost, robustness to malicious clients. Some numerical experiments are performed on these aspects to justify the performance gain.

# Weaknesses
* The authors may ignore some prior works that attempt to introduce the idea of masking to federated learning, like [1].
* The baselines are weak and potentially lead to unfair comparison. Without considering stronger baselines, it is hard to justify the performance gain of the proposed method.
    * A more recent communication-efficient techniques developed for distributed learning need to be considered, e.g., at least the error-feedback framework [2, 3, 4] should be integrated with these compression techniques for FL [10, 11].
    * The paper only considers comparing with FedAvg under non-iid local data distribution. A line of recent studies, e.g. SCAFFOLD [5], FedProx [6], FedNova [9], server Momentum-based FedAvg [7, 8], are ignored. Authors are also encouraged to comment the integrability of these FL methods on the proposed FSL method.
    * It would be great if the authors can provide additional comparison results in terms of different local update updates, number of client participation ratio, local data non-iid-ness, etc.
    * In the appendix, same hyper-parameters are used by all methods. It would be great if the authors can at least provide one Table/Figure to justify that tuning hyper-parameters for all methods will not change the performance gain of the proposed method.


# Reference
1. Dynamic Sampling and Selective Masking for Communication-Efficient Federated Learning, https://arxiv.org/abs/2003.09603, 2020.
2. Error Feedback Fixes SignSGD and other Gradient Compression Schemes, ICML 2019.
3. EF21: A New, Simpler, Theoretically Better, and Practically Faster Error Feedback, NeurIPS 2021.
4. Deep Gradient Compression: Reducing the Communication Bandwidth for Distributed Training, ICLR 2018.
5. SCAFFOLD: Stochastic Controlled Averaging for Federated Learning, ICML 2020.
6. Federated Optimization in Heterogeneous Networks, MLSys 2020.
7. Faster Non-Convex Federated Learning via Global and Local Momentum, 2020.
8. Adaptive Federated Optimization, ICLR 2021.
9. Tackling the Objective Inconsistency Problem in Heterogeneous Federated Optimization, NeurIPS 2020.
10. Bidirectional compression in heterogeneous settings for distributed or federated learning with partial participation: tight convergence guarantees, 2021
11. Linear Convergence in Federated Learning: Tackling Client Heterogeneity and Sparse Gradients, Feb. 2021.

**Summary Of The Paper:**

The paper proposes to collaboratively learn a supermask within a randomly initialized neural networks, instead of learning the model parameters.
This idea is interesting and the authors verify its effectiveness on MNIST, CIFAR and FEMNIST, with the benefits of reduced communication cost and robustness to malicious clients.

**Summary Of The Review:**

In general this paper is interesting; but before the acceptance of the paper, the authors need to provide additional numerical comparison over other strong baselines in the rebuttal (as pointed out in the main review section).

---

> ### Author Response · Authors · 2021-11-17
> **Authors' Response to Reviewer XrNo (part 1)**
>
> Thank you for your helpful feedback. We would like to address your concerns below.
>
> ***
>
> - **C1:** [Re: "The authors may ignore some prior works that attempt to introduce the idea of masking to federated learning, like [1]".]
>
> **Ans1:** [1] combined dynamic client sampling and TopK selective masking to improve communication efficiency. But, unlike FSL, their motivation does not involve providing robustness against poisoning attacks. As explained in Section 6.2, we do not have TopK results in Table 2 as TopK is as vulnerable as vanilla FedAvg, i.e., a single malicious client can reduce the accuracy of TopK to 10% (random guess) for CIFAR10 or MNIST by just sending very large weight parameters.
>
> ***
>
> - **C2:** [Re: "The baselines are weak and potentially lead to unfair comparison"]
>
> **Ans2:** Please note that our main goal of FSL is to provide robustness using a ranking-based FL. Sending and receiving rankings are more robust to adversaries as explained in Section 5.1. To this end, we believe we have considered Multi-Krum and Trimmed-Mean as the two state-of-the-art Byzantine-robust aggregation mechanisms.
>
> ***
>
> - **C3:** [Re: "A more recent communication-efficient techniques developed for distributed learning need to be considered, e.g., at least the error-feedback framework [2, 3, 4] should be integrated with these compression techniques for FL [10, 11]"]
>
> **Ans3:** We believe that we have compared FSL with recent communication efficient techniques including quantization methods (SignSGD) and sparsification methods (TopK). [2,3,4] are introducing error-feedback to distributed learning settings, but to the best of our knowledge, it is not useful to extend them to cross-device FL.  The reason for this is that in these techniques, the workers will save a local state after calculating the gradients to utilize in future rounds to minimize the error (e.g., [4] saves accumulated gradients lower than a threshold or [2,3] save a local version in each round).
> The problem is that in cross-device FL (thousands or millions of devices), only a few clients are chosen for each round. This means that each client may get one chance to participate in the global model update. Therefore, there is no use in saving the local state for future use.
>
> Even if a client is reselected again, the previous local state is not useful anymore since a huge time gap is between these two rounds. For example, in our experiments with FEMNIST, we have 3400 users, and we are choosing 25 of them for each round. If a client is chosen in the round $i^{th}$, then the expected round to be reselected is round $i+136$, where the saved local update for previous participation is no longer useful anymore.
>
> Another problem of cross-device FL is that the clients have small and heterogeneous datasets. For instance, we distribute the datasets (CIFAR10, MNIST) over 1000 clients, so the clients have diverse and small training data (e.g. client 13th in CIFAR10 experiments has data {0: 2, 1: 6, 2: 0, 3: 7, 4: 21, 5: 1, 6: 1, 7: 2, 8: 1, 9: 0} where it shows class: number data points}). Computing a local state does not help these clients to better generalize or reduce their errors in future. On the other hand, in distributed learning settings, each worker has access to more data and also more iid data which make the performance of error-feedback mechanisms better. For example [2] is experimenting using a single node and it did not show the effectiveness for multiple nodes.

---

> > ### Author Response · Authors · 2021-11-17
> > **Authors' Response to Reviewer XrNo (part 2)**
> >
> > - **C4:** [Re: "The paper only considers comparing with FedAvg under non-iid local data distribution. A line of recent studies, e.g. SCAFFOLD [5], FedProx [6], FedNova [9], server Momentum-based FedAvg [7, 8], are ignored. Authors are also encouraged to comment the integrability of these FL methods on the proposed FSL method."]
> >
> > **Ans4:** Our main argument is that FL algorithms based on FedAvg are vulnerable to the same kind of attacks that even a single malicious client can corrupt the model by sending well-crafted updates (e.g. very large updates for FedAvg). This vulnerability comes from sending and receiving weight parameters as the adversary has more space to find the most damaging updates. The clients in SCAFFOLD[5], FedProx[6], FedNova [9], Momentum-based FedAVG [7,8] also send updates as trained weights, which makes them vulnerable to the same attack that FedAvg suffers. On the other hand, FSL uses ranking, and ranks are free-scale which make it more robust compared to these systems. However, for better performance in case of no malicious client, we can integrate their ideas into FSL as follow:
> >
> > - SCAFFOLD [5] estimates the update direction for the global model, and update direction for each client. Then it uses the difference as an estimate of the client drift to correct the local update. The same idea can be applied to FSL to correct the client drifts too. In modified FSL,  each client estimates the global reputation of edges (using global ranking) and local reputation of edges (using local ranking). Finally, the client utilizes the difference of the reputations as the client drift to correct the local ranking.
> >
> > - FedProx [6] tries to learn a local model for a chosen client by regularizing the distance of the local and global model. The same technique can be integrated into FSL too. In modified FSL, each client calculates the local rankings of the supernetwork with this constraint that the local ranking ($R_k$) should be close to the global ranking ($R_g^t$). To measure the difference between two rankings, we can calculate the reputation of each edge using Argsort and compare the difference between reputations of rankings. In this case, we are optimizing the following objective function for $k^{th}$ client ($F_k() is showing the local optimization$):
> >      * in FedProx: $w_k^{t+1} = \arg\min_{w}  F_k(w)+\frac{\mu}{2} ||w-w^t||$
> >      * in modified FSL: $R_{k}^{t+1}=\arg\min_{R} F_k(R)+\frac{\mu}{2} ||\text{Argsort}(R)-\text{Argsort}(R_g^t)||$
> >
> > - FedNova [9] averages the gradients by the number of local epochs that each client has instead of having a fixed number of local epochs per client. The same idea can be integrated into FSL too. In our FSL experiments we used 2 local epochs for MNIST and FEMNIST clients, and 5 local epochs for CIFAR10 users. However, if we use different numbers of local epochs per client, we can use the same technique by averaging the reputation of edges based on the number of local epochs.
> >      * in FedAvg, The server announces the global model as $w^{t+1}=\frac{1}{n} \sum_{i=1}^n W_i^{t}$
> >      * in FedNova, The server announces the global model as $w^{t+1}=\frac{1}{n} \sum_{i=1}^n \frac{W_i^{t}}{\tau_i}$ where $\tau_i$ is the number of local updates for $i^{th}$ client.
> >      * in modified FSL, the server calculates the global rankings by ($T$ is a vector containing the number of the local epoch that each user applied ($\tau_u$) :
> > **def function** Vote($R_{u \in U}, T_{u \in U}$):\
> > &nbsp;&nbsp;	V1=ArgSort($R_{u \in U}$);\
> > &nbsp;&nbsp;	A=Sum(V1/T);\
> > &nbsp;&nbsp;	**return** ArgSort(A)
> >
> > - Momentum-based FSL: [7] uses a global momentum for updating the global model, and [8] expands it to utilizing AdaGrad and Adam. All of these methods can be integrated into FSL too as in the server we are calculating the reputation of each edge and summing them. In momentum-based FSL, the server will incorporate the previous rankings (reputations) for each edge to update the new rankings as momentum.
> > In this case, the function Vote with momentum in Algorithm 2 would be ($\mu$ is the momentum, $R_g^t$ is previous global ranking):
> > **def function**  Vote($R_{u \in U}, R_g^t, \mu$): \
> > &nbsp;&nbsp;	V1=ArgSort($R_{u \in U}$); \
> > &nbsp;&nbsp;	V2=ArgSort($R_g^t$);\
> > &nbsp;&nbsp;	A=Sum(V1+$\mu \times$V2);\
> > &nbsp;&nbsp;	**return** ArgSort(A)
> >
> > ***
> > - **C5:** [Re: "It would be great if the authors can provide additional comparison results in terms of different local update updates, number of client participation ratio, local data non-iid-ness, etc"]
> >
> > **Ans5:** We have added Table 7 (please check the revised version of our submission) in the appendix showing the requested comparisons.

---

> > > ### Author Response · Authors · 2021-11-17
> > > **Authors' Response to Reviewer XrNo (part 3)**
> > >
> > > - **C6:** [Re:In the appendix, the same hyper-parameters are used by all methods. It would be great if the authors can at least provide one Table/Figure to justify that tuning hyper-parameters for all methods will not change the performance gain of the proposed method.]
> > >
> > > **Ans6:** We have added Table 4 in the appendix showing the performance of the FSL when CIFAR10 is distributed over 1000 users and there are 10% malicious clients among them. This table shows that the robustness of FSL still persists (our main goal) even if we change the hyperparameters.
> > >
> > > ***
> > >
> > > In Conclusion, the goal of FSL is performing federated learning by using ranks instead of training weights, and since ranks are scale-free, they are providing stronger robustness (as demonstrated both analytically and through experiments). We have compared FSL with the state-of-the-art robustness mechanisms,  Multi-Krum and Trimmed-Mean, to show its effectiveness. Even these mechanisms have access to the number of malicious clients in each round, while FSL does not need this information as it provides robustness by its design (using ranks).

---

> ### Comment · Reviewer_XrNo · 2021-11-17
> **reply to the authors' response**
>
> Thanks for providing the responses as well as the additional results.
>
> I think in the abstract, the submission is motivated by the statement "In-the-wild deployment of FL faces two major hurdles: robustness to poisoning attacks and communication efficiency." When reading the current responses, it seems like the authors have changed the tongue and argued that "our main goal of FSL is to provide robustness using a ranking-based FL".
>
> I agree with the argument regarding the cross-device setting, where stateless clients may only appear once during the training. However, authors need to provide empirical evidence for its justification, e.g., (1) the statement like "there is no use in saving the local state for future use" for stateless client scenario, (2) the statement like "the saved local update for previous participation is no longer useful anymore". Unless the authors can provide some evidences, otherwise these statements cannot fully convince me.
>
> Regarding the practicability of the error feedback framework, I think its effectiveness has been verified either by the theoretical work (provided in the initial review), or other empirical works, e.g. [Ref1, Ref2].
>
> Authors also argue that "the ingredient of some recent FL work can be integrated into FSL", it would be great if some numerical evidences can be provided.
>
> * [Ref1] Deep Gradient Compression: Reducing the Communication Bandwidth for Distributed Training, ICLR 2018.
> * [Ref2] PowerSGD, a default gradient compressor introduced in Pytorch 1.8+. It also relies the error feedback framework.

---

> > ### Author Response · Authors · 2021-11-29
> > **Authors' Response to Reviewer XrNo (part1)**
> >
> > Thank you for your helpful feedback. We would like to address your concerns below.
> >
> > -----
> > - **CC1:** "... When reading the current responses, it seems like the authors have changed the tongue and argued that "our main goal of FSL is to provide robustness using a ranking-based FL".
> >
> > **AA1:** The main objective of FSL is improving robustness, for which we provide experimental and analytical evaluations. As the approach we take relies on rankings, a side-advantage of FSL is also improving communication efficiency; however, improving communication efficiency is not our main objective, as there are other solutions that specifically target communication efficiency with better performances.
> >
> > -----
> > - **CC2:** "I agree with the argument regarding the cross-device setting, where stateless clients may only appear once during the training. However, authors need to provide empirical evidence for its justification, ..."
> >
> > **AA2:** Peter Kairouz et al. [RefA: page 33, paragraph 2] also emphasized: "In cross-device FL, algorithms generally cannot assume any state is preserved on the clients. However, this constraint would typically not be present in the cross-silo FL setting, where the same clients participate repeatedly". They also mentioned [RefA: page 6, Table 1]: "Cross-device FL is Stateless — each client will likely participate only once in a task, so generally a fresh sample of never-before-seen clients in each round of computation is assumed".
> >
> > [RefB] states that the problem with SignSGD is that the sign operator loses (i.e. forgets) information about (I) magnitude and (II) direction of the gradients. Therefore they suggest that the clients scale the compressed gradients by the magnitude of the actual gradient to prevent losing information about the magnitude. They also suggest saving the difference of the actual gradient and scaled compressed gradient as an error locally, and adding this to the next round to prevent losing information about the direction.
> >
> > To show validation of our statements, we consider our cross-device setting for CIFAR10. In the following table, we measure the magnitude and direction change of 25 random clients' updates in the 200th, 201st, and 400th global rounds when there is no malicious attacker or 10% malicious clients. we consider 200th to 201st to show the effectiveness of saving local state if the client participates in all rounds, and 200th-400th to show cross-device setting that there is a gap between two participation.
> >
> > | rounds   |      average magnitude difference when there is no attacker      |  average direction change when there is no attacker |   average magnitude difference when there is 10% malicious clients      |  average direction change when there is 10% malicious clients|
> > |----------|:-------------:|------:|:-------------:|:------:|
> > | round200 to round201 |  0.225 | 34.8% |  1.192 | 53.6% |
> > | round200 to round400 |    0.874   |   48.8% |  1.973 | 47.2% |
> >
> > In this table, we can see if the client can keep its state for each round to apply for the next round (round200 to round201), the difference is small and it can compensate for the error, but when the gap between two rounds is large, the previous state is totally different and not helpful anymore. When we have poisoning clients in the system, the difference would be even larger. We even integrated [RefB] into our SignSGD experiments, and the accuracy of the clients was reduced from 79.1% to 34.7%, which validates our statements.
> >
> > [RefA]: Kairouz, Peter, et al. "Advances and open problems in federated learning." arXiv preprint arXiv:1912.04977 (2019).
> >
> > [RefB]: Karimireddy, Praneeth, et al. "Error Feedback Fixes SignSGD and other Gradient Compression Schemes", ICML 2019.

---

> > > ### Author Response · Authors · 2021-11-29
> > > **Authors' Response to Reviewer XrNo (part2)**
> > >
> > > - **CC3:** "Regarding the practicability of the error feedback framework, I think its effectiveness has been verified either by the theoretical work (provided in the initial review), or other empirical works, e.g. [Ref1, Ref2]".
> > >
> > > **AA3:** We believe keeping the state of the updates in distributed learning and cross-silo FL is useful to compensate for the errors. However, in our experiment we consider cross-device FL, where the number of nodes is much larger, and the probability that a single client participates multiple times in training is very low [RefA: page 6, Table 1]], so keeping the state of previous training is of little use. Note that this probability drops quickly as the number of nodes/clients increases which is the case in production cross-device FL systems.
> > >
> > > [Ref1] is using a single node (or 4 nodes) in most of its experiments that all of them are participating in all the training rounds. [Ref2 paper] used 8 nodes that all of them are participating in all the training rounds. [Ref2 Pytorch version] is also designed for distributed learning where all the clients are participating in updating the global model.
> > >
> > > [RefA]: Kairouz, Peter, et al. "Advances and open problems in federated learning." arXiv preprint arXiv:1912.04977 (2019)
> > >
> > > ----
> > >
> > > - **CC4:** "Authors also argue that "the ingredient of some recent FL work can be integrated into FSL", it would be great if some numerical evidences can be provided."
> > >
> > > **AA4:** Due to limited time, we provide results for the momentum FSL, but we can provide results for the other FSL variants in the final version.  We measure the performance of integrating a global momentum into FSL. The mean of accuracies of all the 1000 clients after 500 global rounds is changed from 51.3% to 54.3% which is showing that using a global momentum can help FSL to converge faster.
> > >
> > > In the end, we want to reemphasize that to the best of our knowledge, FSL is the first FL framework working on local and global rankings compared to traditional FL frameworks that use weight training. The clients in SCAFFOLD[5], FedProx[6], FedNova [9], Momentum-based FedAVG [7,8] send updates as trained weights, which makes them vulnerable to the same attack that FedAvg suffers. While in FSL, the clients are training a global ranking which makes it more robust to poisoning attackers.

---

> > > ### Comment · Reviewer_XrNo · 2021-11-29
> > > **additional feedback**
> > >
> > > Thank for providing the additional results and justifications. I am willing to raise my score to 6 since most of my concerns have been addressed. Authors are also encouraged to refactor the arguments/statements in the draft later.
> > >
> > > One followup comments (clarification is needed): why the magnitude and direction change of 25 random clients' updates in the 200th, 201st, and 400th global rounds are valuable metrics to indicate the usefulness of the error-feedback framework for the cross-device stateless clients? When talking about the 'the accuracy of the clients was reduced from 79.1% to 34.7%', is it refer to the setup and result in Table 1, CIFAR10 + Conv8+1000 clients, where the stale local feedback memory only has negative effects?

---

> > > > ### Author Response · Authors · 2021-11-29
> > > > **Thank you for your feedback**
> > > >
> > > > Thank you for all the comments. We will refactor all the arguments/statements in the final version.
> > > >
> > > > Yes, the accuracy is the result of singSGD+error-feedback+Conv8+1000clients, but we did not tune the hyper-parameters for this experiment, and we use the original hyperparameters we used for signsgd (without EF) experiments."

---

> > > > > ### Comment · Reviewer_XrNo · 2021-11-29
> > > > > **need clarification**
> > > > >
> > > > > Thanks for the reply.
> > > > >
> > > > > Authors are encouraged to first properly address my left main concern. It would be great if the authors could
> > > > > 1. clarify why the magnitude and direction change of 25 random clients' updates in the 200th, 201st, and 400th global rounds are valuable metrics to indicate the usefulness of the error-feedback framework for the cross-device stateless clients?
> > > > > 2. tune the hyper-parameter of signSGD+EF. Such a big drop caused by stale local memory is quite surprising to me: it is unfair to directly use the optimal learning rate of signSGD for signSGD+EF, and it is unclear where the quality loss comes from. Based on my own experiences of gradient compression for non-FL scenarios, signSGD mimics the behaviours of Adam and thus Adam's optimal learning rate, while signSGD+EF indeed recovers the behaviours of SGD; as a result, the magnitude of optimal learning rate in signSGD normally is at least 10-100 times smaller than signSGD+EF.
> > > > >
> > > > > Given the no response to the first point and unfair comparison mentioned in the second point, even though I am somehow convinced by the limitation of error-feedback framework for cross-device setup, some strong and correct numerical evidences should be provided (as the current experiments only consider the low client participation case, instead of the real stateless case).

---

> > > > > > ### Author Response · Authors · 2021-11-30
> > > > > > **Authors' Response to Clarification**
> > > > > >
> > > > > > Thank you for your reply. We understand your concerns and we tried to address them below:
> > > > > >
> > > > > > -----
> > > > > >
> > > > > > 1) We would like to address your first concern below:
> > > > > >
> > > > > > - **a) Why we measured the difference between two rounds (200th to 201st and 200th to 400th).**
> > > > > >
> > > > > > We considered these two round intervals to show the following:
> > > > > >
> > > > > > -- I) the difference between two sequential rounds (200th to 201st): in this case, we are measuring the metrics in a distributed-learning/cross-silo-FL setting as the same clients have the right to participate in all the rounds (every sequential round).
> > > > > >
> > > > > > -- II) the difference between two rounds with a large gap (200th to 400th): in this case, we are measuring the metrics in a cross-device setting where there is a large time gap between two rounds of participation.
> > > > > >
> > > > > > - **b) Why we measured the magnitude and direction change of the updates in our SignSGD+EF to indicate the usefulness of the error-feedback framework for the cross-device stateless clients.**
> > > > > >
> > > > > > [RefB] states that the problem of the sign operator in SignSGD is that it will forget information about the magnitude and direction of the gradients. Therefore, the authors add two mechanisms to compensate for this information loss:
> > > > > >
> > > > > > -- I) to reduce forgetting about the magnitude: scale the sign gradient by the norm of the actual gradient
> > > > > >
> > > > > > -- II) to reduce forgetting about the direction of gradient: save the difference between the actual gradient and scaled sign gradient and add it in the next rounds.
> > > > > >
> > > > > > Now, in our experiment, we tried to show that EF is not as useful as in cross-device-FL compared to cross-silo-FL/distributed-learning settings. In the cross-device-FL, the magnitude difference is 4 times the magnitude difference in cross-silo-FL/distributed-learning. Also around 50% more parameters are moving in another direction (changed from + to - or - to +) in cross-device-FL compared to cross-silo-FL/distributed-learning.
> > > > > >
> > > > > > EF can compensate for the error in cross-silo-FL/distributed-learning as the difference between two sequential rounds is not much, but in cross-device-FL, the difference is much more and the previous state of the client cannot compensate for the error, and even it can corrupt it.
> > > > > >
> > > > > > - **c) why 25 clients?**
> > > > > > We reported the average magnitude and number of direction changes of the 25 clients. At first, we picked 25 clients among 1000 clients randomly, but then we keep them fixed for the measurements. We picked 25 since we have just enough memory to save this number of updates. However, these are a sample of all the clients showing that the EF is not useful for these 25 clients.
> > > > > >
> > > > > >
> > > > > > ------
> > > > > >
> > > > > > 2) **Potential reasons why we have a huge drop in the accuracy:**
> > > > > >
> > > > > >
> > > > > > - a) Please note that the original SignSGD was proposed for FedSGD where signs are communicated after every gradient-step, but due to its efficacy in FedAvg, it is also used in the FL setting where signs are communicated after E local epochs. For instance, we observe that E=5 is the best for the FedAvg variant of SignSGD in CIFAR10. One reason that we see such a drop in the accuracy could be that the saved local errors after E local epochs are not helpful as the original per-gradient errors are.
> > > > > >
> > > > > > - b) In the first experiment, we show that the saved local error for a long time ago (cross-device-FL) is not useful as the current update of the client is much different from its last update.  In cross-device FL, we sample a small portion of the clients in each round randomly to calculate their local updates, and for some clients, the gap between two selection rounds would be much larger that it can corrupt their future model updates result in accuracy reduction.
> > > > > > ----
> > > > > >
> > > > > > We want to reemphasize that FSL's first goal is to provide robustness against poisoning attackers. A poisoning attacker can poison the global model and the local states in an FL framework even with Error-Feedback as when the original model has poisoned the local training on a poisoned global model results in a poisoned local state too. Today is the last day of rebuttal, and we do not have enough time to tune the parameters for singSGD+EF for cross-device settings. **Given more time, we can provide more results in the final draft**.

---

### Official Review · Reviewer_bj9t · 2021-11-02

**Correctness:** 2
**Technical Novelty And Significance:** 2
**Empirical Novelty And Significance:** 2
**Recommendation:** 3
**Confidence:** 4

**Main Review:**

Strength

1. A federated supermask learning is proposed for improving the communication efficiency of federated learning

2. The edge ranking communication improves the efficiency of the federated supermask learning itself.

3. Extensive evaluation on three real datasets demonstrate the superior performance of the proposed approach.

Weakness

1. The authors leverage the EdgePop Algorithm [2] for model pruning. The original paper is only able to prove that switching edges according to iterated edge scores can reduce the loss on randomly initialized networks. The connection between the EdgePop Algorithm and the model pruning remains unclear.

2. This paper is essentially a pruning-before-training algorithm. However, pruning-after-training and pruning-during-training techniques often achieve the better approximation. More detailed analysis are expected to explain the motivation and benefits of the choice, and whether the proposed method is able to guarantee the model accuracy.

3. Utilizing network pruning to accelerate the FL training for reducing the communication cost is not a new research problem [1]. The authors should discuss recent advances in federated learning. In addition, the authors fail to discuss and compare with existing pruning-before-training techniques.

4. The paper lacks of enough novelties, since two main components of the paper, EdgePop and voting, are from existing techniques.

[1] Yuang Jiang et al. "Model Pruning Enables Efficient Federated Learning on Edge Devices". In: CoRR abs/1909.12326 (2019).

[2] Vivek Ramanujan et al. "What's Hidden in a Randomly Weighted Neural Network?" In: 2020 IEEE/CVF Conference on Computer Vision and Pattern Recognition, CVPR 2020, Seattle, WA, USA, June 13-19, 2020. Computer Vision Foundation / IEEE, 2020, pp. 11890–11899.

**Summary Of The Paper:**

This paper presents a pruning-before-training algorithm for improving the communication efficiency of federated learning. In each iteration, all local devices with the same initial weights perform the pruning on their own local datasets. The edge scores produced by local devices are aggregated by the server. The order of the edge scores will be used in the next iteration of score initialization process. After the pruning convergence, the edges are pruned correspondingly and the federated learning model is trained on the pruned network.

**Summary Of The Review:**

Overall, the well designed structure makes the workﬂow clear and easy to follow, but the further analysis and discussion are expected to clarify the contributions in the techniques as well as in the evaluation section.

---

> ### Author Response · Authors · 2021-11-16
> **Authors' Response to Reviewer bj9t**
>
> Thank you for your helpful feedback. We would like to address your concerns below.
>
> ***
>
> - **C1:** [Re: “The authors leverage the EdgePop Algorithm [2] for model pruning. The original paper is only able to prove that switching edges according to iterated edge scores can reduce the loss on randomly initialized networks. The connection between the EdgePop Algorithm and the model pruning remains unclear.”]
>
> **Ans1:** Note that, FSL is orthogonal to the techniques that combine pruning and training because FSL never trains the weights, and instead, only searches for a subnetwork within an initial randomly initialized neural network. FSL does not prune the network during training, instead, it finds a global ranking of edges in the network without changing its weights (Algorithm 2).
>
> ***
>
> - **C2:** [Re: "This paper is essentially a pruning-before-training algorithm. However, pruning-after-training and pruning-during-training techniques often achieve the better approximation. More detailed analysis are expected to explain the motivation and benefits of the choice, and whether the proposed method is able to guarantee the model accuracy."]
>
> **Ans2:** As described in Section 4, in each FSL round, FSL clients collaborate to learn a global ranking of the edges in a neural network whose weights are randomly initialized and kept unchanged. Again, we want to reemphasize that FSL is not a pruning-before-training, and there is no weight training in FSL.
>
> As detailed in Section 5.1, along with communication efficiency, FSL provides robustness because rankings reduce the degrees of freedom for adversaries to craft malicious updates. Using pruning-before-training, pruning-after-training and pruning-during-training involve the step of updating the weights and sending them to the server for aggregation. This step makes all of them vulnerable as the malicious clients can send very large parameters even for a pruned subnetwork. On the other hand, FSL does not train weights as it is calculating local and global rankings.
>
> ***
>
> - **C3:** [Re: "Utilizing network pruning to accelerate the FL training for reducing the communication cost is not a new research problem [1]. The authors should discuss recent advances in federated learning. In addition, the authors fail to discuss and compare with existing pruning-before-training techniques."]
>
> **Ans3:** FSL is not about "Utilizing network pruning to accelerate the FL training". We do not prune the network to accelerate the training as we do not train the weights, we keep them fixed through all the FSL rounds.
>
> On the other hand, PruneFL [1] is a pruning-before-training by training weights multiple times using SGD and removing a certain percentage (referred to as the pruning rate) of the weights that have smallest values layer-wise. This training and pruning process is repeated until a desired size of trained subnetwork. And at the end, each client sends the trained subnetwork to the server for aggregation. We argue that training weights (as any other pruning-before-training methods) makes it vulnerable as a malicious client can send very well-crafted weights (e.g. very large parameters) even for an aggressively pruned subnetwork, and jeopardize the global model completely since the server will average the submitted weights. However, in FSL, we do not train weight parameters as FSL clients are learning local ranking to combine and generate a global ranking of fixed weights.
>
> On top of that,  we already talked about LotteryFL[a] which combines the lottery ticket hypothesis and Fedvg as a pruning-before-training in Section E. In this section, we explained the difference between FSL LotteryFL (an example of pruning-before-training).
>
> [a] Ang Li et al. "Lotteryfl: Personalized  and  communication-efficient  federated  learning  with  lottery  ticket  hypothesis on non-iid datasets". https://arxiv.org/abs/2008.03371
>
> ***
>
> - **C4:** [Re: "The paper lacks of enough novelties, since two main components of the paper, EdgePop and voting, are from existing techniques."]
>
> **Ans4:** We believe that to the best of our knowledge, ours is the first work to use combination of voting and supermasks to give provably robust and communication efficient FL algorithm. To best our knowledge there is no existing FL using a ranking voting mechanism.
>
> ***
>
> In conclusion, we would like to emphasize that the motivation of FSL is to learn rankings (which are free-scale) instead of learning weights (what we have in pruning-before-training). Learning local and global rankings make the FSL robust as we showed theoretically and empirically. In this paper, we showed that FL systems based on weight training (such as FedAvg, Byzantine-robust AGRs, TopK, and LotteryFL) are more vulnerable to poisoning attacks as the adversary has more freedom to generate well-crafted weight parameters.

---

> > ### Comment · Reviewer_bj9t · 2021-11-22
> > **Reply to the authors' response**
> >
> > Thanks for the authors' detailed discussion and explanation! After reading the authors' responses, most of raised concerns and issues are still to be addressed.
> >
> > This paper is essentially a model pruning algorithm. As the authors claimed in the paper, "FSL server trains a global subnetwork within a randomly initialized neural network (i.e., supernetwork)". "The subnetwork is then used for downstream tasks, e.g., image classification". Thus, the generated subnetwork can be treated as a pruned version.
> >
> > Many existing pruning-before-training algorithms directly prune the neural networks without weight training.
> >
> > In addition, the paper claims "by sharing subnetworks, FSL reduces the communication cost of training". Reducing the communication cost will not accelerate the training?

---

> > > ### Author Response · Authors · 2021-11-29
> > > **Authors' Response to Reviewer bj9t**
> > >
> > > Thank you for your helpful feedback. We would like to address your concerns below.
> > >
> > > -----
> > >
> > > - **CC1:** "FSL is a pruning algorithm."
> > >
> > > **AA1:** We want to emphasize that we use Edge-Popup (a pruning algorithm) to generate rankings in FSL. We believe that in a pruning based FL [a-e], the goal of the clients is to prune the network to reduce the number of parameters communicated between server and clients, but these works do not empirically/theoretically analyze robustness of their proposals. On the other hand, in FSL, each client sends and receives all the local and global rankings related to all of the parameters. We even consider a pruned version of FSL in section 5.2 called *Sparse-FSL* (SFSL) in which the clients just send a portion of rankings (rank of important edges) to the server to reduce the upload bandwidth.
> > >
> > > Please note that our main contribution is to demonstrate that a ranking based FL is more robust compared to existing FL systems based on weight training (even when such they are combined with weight pruning). For example, in following pruning methods in FL [a-e], even one malicious client can jeopardize the global model by sending very large updates for its pruned subnetwork. However, FSL is robust by design against such and other state-of-the-art untargeted poisoning attacks, as we show theoretically (section 5.1 and A.2) and empirically (Section 6.2).
> > >
> > > [a] Ang Li et al. "Lotteryfl: Personalized and communication-efficient federated learning with lottery ticket hypothesis on non-iid datasets". [https://arxiv.org/abs/2008.03371]
> > >
> > > [b] Yuang Jiang et al. "Model Pruning Enables Efficient Federated Learning on Edge Devices". In: CoRR abs/1909.12326 (2019).
> > >
> > > [c] Dynamic Sampling and Selective Masking for Communication-Efficient Federated Learning, [https://arxiv.org/abs/2003.09603], 2020.
> > >
> > > [d] Dan Aistarh et al. "The convergence of sparsified gradient methods". NeurIPS, 2018.
> > >
> > > [e] Alham Fikri Aji et al. "Sparse communication for distributed gradient descent." EMNLP, 2017.
> > >
> > > ----
> > >
> > > - **CC2:** "Many existing pruning-before-training algorithms directly prune the neural networks without weight training."
> > >
> > > **AA2:** To the best of our knowledge, there is no existing federated learning based on ranking (i.e. without training weights and just based on pruning-before-training algorithms). We believe FSL is more than simply pruning trained or untrained network, because:
> > > 1) we use the Edge-popup to create local rankings, (Figure 1 - Step 1)
> > > 2) we provide a novel voting mechanism (Figure 1 - Step 3) to combine local rankings into a single global ranking,
> > > 3) we design local rank training (Figure 1 - Step 2a) by how each client is starting from a global ranking and train its own local rankings for the next training round.
> > > 4) Finally, we provide both empirical and theoretical robustness analyses of our proposal.
> > >
> > > ----
> > >
> > > - **CC3:** "In addition, the paper claims "by sharing subnetworks, FSL reduces the communication cost of training". Reducing the communication cost will not accelerate the training?"
> > >
> > > **AA3:** By "by sharing subnetworks, FSL reduces the communication cost of training", we mean that by using FSL we can create rankings, and sending and receiving rankings allow FSL to reduce the communication cost. This may accelerate training, but understanding the convergence properties of FSL is out of the scope of this work.

---

### Official Review · Reviewer_q5LC · 2021-11-02

**Correctness:** 3
**Technical Novelty And Significance:** 2
**Empirical Novelty And Significance:** 2
**Recommendation:** 6
**Confidence:** 5

**Main Review:**

Strengths:
1)	This paper combines “supermask” techniques with FL and improve the communication efficiency and robustness of FL at the same time.
2)	This paper provides thorough empirical results to support its claim about the improvement of communication efficiency and robustness.
Weaknesses:
1)	The experimental settings are not hard enough to evaluate the performance of FSL. There is no doubt that there is information loss when the devices transmit only the ranking of scores. This kind of information loss is not serious when the rankings on different devices are similar (the local subnetwork structures are similar due to similar data distributions). In this paper, the user uses Dirichlet distributions to construct non-IID data for MNIST and CIFAR10. Even though the data distributions are different across devices, each device still holds all classes of data and local subnetwork structures would not show significant difference. And I guess the robustness results have the same problem since the authors use a voting mechanism to update the global ranking. I am wondering whether FSL would perform well under the non-IID setting in FedAvg paper, where each client only has two classes of data rather than all classes of data.
2)	The improvement over baselines is not significant on some dataset. For example, “Top-K 10%” achieves even higher accuracy than “FSL” with lower communication cost on FEMNIST dataset.
3)	The idea of utilizing “supermask” seems novel, but this paper seems just simply combining “supermask” with FL. It is okey to do “A plus B” things, but you need to provide some scientific contributions like providing a theoretical analysis about why “supermask plus FL” works, and what challenges that you solved make it deserve an acceptance by a top avenue like ICLR.


**Summary Of The Paper:**

This paper exploit the "supermask" technique to improve both robustness and communication efficiency in federated learning. The collaborating clients only share the local subnetwork based on the rankings of network edges, and the server performs aggregations over the local subnetworks of all participating clients. The authors also provides theoretical and empirical study on how the proposed method can improve robustness and communication efficiency.

**Summary Of The Review:**

This paper utilizes “supermask” technique in FL to improve communication efficiency and robustness simultaneously and provides thorough empirical studies. However, it seems simply combines “supermask” and FL without solving any challenges nor providing theoretical analysis about why this can work. In addition, the experimental settings are not hard enough to judge the usability of the algorithm proposed by this paper. Based on the experimental settings and the lack of scientific contributions, I would give a negative score. If the authors can solve these two major concerns, I am delighted to increase my score.

---

> ### Author Response · Authors · 2021-11-16
> **Authors' Response to Reviewer q5LC**
>
> Thank you for your helpful feedback. We would like to address your concerns below.
>
> ***
>
> - **C1:** [Re: "...  I am wondering whether FSL would perform well under the non-IID setting in FedAvg paper, where each client only has two classes of data rather than all classes of data."]
>
> **Ans1:** We have added Table 6 (please check the revised version of our submission) showing the effect of two different data assignments on the performance of FSL and FedAvg. We also added notebooks in the supplementary materials showing that FSL works with the data distribution of 2 random classes assigned to each client similar to the original FedAvg. We explained how we distribute the heterogeneous data in section C.3 in the appendix.
>
>
> Please also note that, even in non-iid data distribution using Dirichlet, all clients do not necessarily have samples from all the classes. For example client 13th in CIFAR10 experiments has data {0: 2, 1: 6, 2: 0, 3: 7, 4: 21, 5: 1, 6: 1, 7: 2, 8: 1, 9: 0} where it shows class:number data points} that it does not have any samples from class 2 and 9. Also, the number of data points for class 4 is much more than others which makes it more realistic in real-world scenarios. We printed these details of data assignments in our notebooks (supplementary materials).
>
> ***
>
> - **C2:** [Re: ”The improvement over baselines is not significant on some dataset. For example, “Top-K 10%” achieves even higher accuracy than “FSL” with lower communication cost on FEMNIST dataset.”]
>
> **Ans2:** Note that TopK has better performance *only* in the absence of any poisoning attacks (and only for FEMNIST data). Our goal in this paper is to resist poisoning attacks. As mentioned in Section 6.2, TopK is as susceptible as vanilla FedAvg to poisoning attacks, i.e., even a single malicious client can jeopardize the accuracy of the global model by crafting very large updates (for x% of parameters). On the other hand, FSL is highly robust to poisoning attacks as it is based on rankings rather than model parameters. Hence, FSL provides high robustness at a very negligible (only for FEMNIST) cost to performance in the benign setting (without any attacks). We attribute this robustness to our ranking algorithm as detailed in Section 5.1.
>
> ***
>
> - **C3:** [Re: ”... provide some scientific contributions like providing a theoretical analysis about why “supermask plus FL” works, and what challenges that you solved make it deserve an acceptance.”]
>
> **Ans3:** Please note that, as we mentioned in the Introduction, a major part of our motivation is to design a robust FL algorithm that is also communication efficient. To this end, we already have theoretical guarantees of the robustness of our voting algorithm in Section 5.1 and A.2. We also present the theoretical basis for the performance of FSL by showing that if two edges are swapped by their rankings in FSL, then the loss of FSL decreases (Theorem 1 in Section D).
>
> ***
>
> - **C4:** [Re: "The idea of utilizing “supermask” seems novel, but this paper seems just simply combining “supermask” with FL."]
>
> **Ans4:** The Edge-Popup paper [a] designed supermask training by assigning scores to different edges, and training them in a randomly neural network. However, FSL is not just about training scores, it is about a ranking-based FL. To the best of our knowledge, there is no FL system that utilizes global and local rankings to learn a global task. The main challenge that we solved in this paper is a novel voting mechanism to combine the local rankings into a global ranking using the reputations of the edges.
>
> [a] Vivek Ramanujan et al. "What's Hidden in a Randomly Weighted Neural Network?",CVPR 2020

---

### Decision · Program_Chairs · 2022-01-20

**Decision:**

Reject

**Comment:**

The paper brings the "supermask" idea used in neural architecture search to the application of federated learning, here represented by a single mask of a larger network. The method can be seen as pruning-before-training, or more precisely pruning-instead-of-training. It is a simplified version of the related works LotteryFL, PruneFL or FedMask, with the difference that here no personalization and no training of the weights is performed, only learning of a global mask. Related work discussion should be improved. While the communication efficiency impact of the method seems minor but positive, the interesting point is that authors here argue that masking will improve robustness to adversarial participants during training.

Unfortunately no theoretical evidence is provided for success of training, in the sense of Byzantine robustness. It is known that robust training can be attacked with small perturbations correlated over time (e.g. 'little is enough'), so also over layers, two important aspects which are ignored here - as voting here is only analysed static at a single time-point. As pointed out by reviewer JJjz, the considered attack (inverting ranking) is far from being formally proven to be the strongest one, and we would have wished for a more precise discussion of these issues as the target of the paper seems to be mainly robustness.

Concerns on the paper also remained on the level of novelty, as it only uses existing building blocks which are more or less directly applicable from the centralized setting, and on the limited contributions towards formal robustness, and on the limited discussion of related work mentioned by several reviewers, only some of which we were able to address in the discussion phase.

We hope the detailed feedback helps to strengthen the paper in the future.